# Regulators of rDNA array morphology in fission yeast

**Alexandria J. Cockrell**[1,2], **Jeffrey J. Lange**[1], **Christopher Wood**[1], **Mark Mattingly**[1], **Scott M. McCroskey**[1], **William D. Bradford**[1], **Juliana Conkright-Fincham**[1,3], **Lauren Weems**[1], **Monica S. Guo**[4], **Jennifer L. Gerton**[1,2]*

1 Stowers Institute for Medical Research, Kansas City, Missouri, United States of America, 2 Department of Biochemistry and Molecular Biology, University of Kansas Medical Center, Kansas City, Kansas, United States of America, 3 Promega Corporation, Madison, Wisconsin, United States of America, 4 Department of Microbiology, University of Washington School of Medicine, Seattle, state of Washington, United States of America

* jeg@stowers.org

**Data Availability Statement:** Original data underlying this manuscript can be accessed from the Stowers Original Data Repository at http://www.stowers.org/research/publications/libpb-2399.

## Abstract

Nucleolar morphology is a well-established indicator of ribosome biogenesis activity that has served as the foundation of many screens investigating ribosome production. Missing from this field of study is a broad-scale investigation of the regulation of ribosomal DNA morphology, despite the essential role of rRNA gene transcription in modulating ribosome output. We hypothesized that the morphology of rDNA arrays reflects ribosome biogenesis activity. We established GapR-GFP, a prokaryotic DNA-binding protein that recognizes transcriptionally-induced overtwisted DNA, as a live visual fluorescent marker for quantitative analysis of rDNA organization in *Schizosaccharomyces pombe*. We found that the morphology—which we refer to as spatial organization—of the rDNA arrays is dynamic throughout the cell cycle, under glucose starvation, RNA pol I inhibition, and TOR activation. Screening the haploid *S. pombe* Bioneer deletion collection for spatial organization phenotypes revealed large ribosomal protein (RPL) gene deletions that alter rDNA organization. Further work revealed RPL gene deletion mutants with altered rDNA organization also demonstrate resistance to the TOR inhibitor Torin1. A genetic analysis of signaling pathways essential for this resistance phenotype implicated many factors including a conserved MAPK, Pmk1, previously linked to extracellular stress responses. We propose RPL gene deletion triggers altered rDNA morphology due to compensatory changes in ribosome biogenesis via multiple signaling pathways, and we further suggest compensatory responses may contribute to human diseases such as ribosomopathies. Altogether, GapR-GFP is a powerful tool for live visual reporting on rDNA morphology under myriad conditions.

## Author summary

Cells devote significant energy and resources to produce ribosomes, large protein complexes that carry out protein synthesis. To ensure that ribosome production matches cellular needs, activation of ribosome assembly is regulated by many inputs. One key

**Funding:** This work was supported by funding from the Stowers Institute https://www.stowers.org/ to JLG. AJC was supported by a training grant F30GM140716 through the NIH-NIGMS https://grants.nih.gov/. MSG was supported by R00-GM134153 from the NIH https://grants.nih.gov/. JLG, AJC, and MSG received salary support from their respective funders. Stowers provided salary support for all other authors. The funders had no role in study design, data collection and analysis, decision to publish, or preparation of the manuscript.

**Competing interests:** The authors have declared that no competing interests exist.

regulatory target is transcription of the tandemly repeated rRNA genes in the ribosomal DNA arrays. A broad understanding of how cells regulate the organization of the rDNA under different cellular conditions and transcriptional states has been difficult to assess with current tools and model systems. We present a new imaging reporter, a fluorescently tagged DNA-binding protein called GapR, that highlights rDNA spatial organization in fission yeast, a model system that shares similar regulatory networks with mammals. Using GapR, we found that the 3D organization of the rDNA arrays changes with the cell cycle, glucose starvation, RNA pol I inhibition, and TOR signaling. We surveyed hundreds of yeast mutants to identify factors that regulate rDNA spatial organization, discovering that the deletion of ribosome protein genes causes rDNA array expansion. Further experiments indicated that ribosome protein gene insufficiency activates conserved regulatory signaling pathways as part of a compensatory response. Our work presents the first broad investigation for regulators of rDNA spatial organization in any organism. Importantly, we identify a compensatory response to compromised ribosome biogenesis that could be conserved in mammals.

## Introduction

Ribosome production is an energy-intensive process that requires strict regulation of several assembly steps. Within the nucleolus, a key step of ribosome biogenesis is transcription of the rRNA genes in the rDNA arrays. The rRNA genes encode the four ribosomal RNAs (rRNAs) found in mature ribosomes, with three of these (18S, 5.8S, 28S) being transcribed by RNA polymerase (pol) I. rRNA transcripts comprise half or more of all transcripts within a cell, a feat that is supported by the simultaneous transcription of multiple rRNA gene repeats [1,2]. To ensure that ribosome biogenesis output matches cellular needs, a multitude of cell signaling pathways regulate RNA pol I activity in response to environmental cues, stressors, and nutrient availability. In cancer, rRNA production is upregulated to support cellular proliferation [3]. RNA pol I activity is a promising therapeutic target for several cancers, necessitating a broad understanding of the factors that regulate the dynamic organization of rDNA to modulate transcription [4].

Many cell signaling pathways that modulate ribosome biogenesis are conserved between mammals and fission yeast. The TOR and cAMP pathways upregulate ribosome synthesis in response to nutrient availability [5,6]. Two conserved MAPK pathways modulate ribosome output in response to stress. These include the Stress-Activated MAPK pathway, orthologous to the mammalian p38 pathway, and the Cell Integrity Pathway, the suspected ERK ortholog for its role in responding to extracellular stress [7,8]. The repressive PI3K/PTEN pathway has conserved orthologs in fission yeast, although their role in yeast cell function is unclear [9]. Given the significance of ribosome synthesis for cellular proliferation, many oncogenes and tumor suppressors influence these regulatory pathways. Cell signaling activity in cancer and in stress that impacts ribosome biogenesis is still under investigation but is likely to be conserved across eukaryotes and could impact the organization of the rDNA.

Changes in ribosome biogenesis are reflected in the structural organization of the nucleolus. Nucleolar hypertrophy serves as a prognostic marker for many cancers, and studies have analyzed nucleolar structure, number, and output to identify novel regulators of ribosome biogenesis [3,10–13]. Underlying nucleolar morphology is the spatial organization of the rDNA arrays themselves, whose organization has been difficult to assess. How rDNA spatial organization is coupled to or distinct from nucleolar structure is unclear. In budding yeast, the

rDNA locus demonstrates variable organization during the cell cycle despite constant nucleolar structure [14]. Transcriptional inactivation is known to alter both nucleolar and rDNA organization in both budding yeast and mammalian cells. Nutrient starvation and direct RNA pol I inhibition cause nucleolar condensation as well as rDNA compaction and reorientation to the nucleolar periphery [15,16]. A thorough investigation of factors that regulate rDNA spatial organization–as well as an understanding of how this spatial organization reflects function–has not been conducted, in part due to limitations of existing approaches to visualize the rDNA. In fission yeast these approaches have been limited to FISH and DNA-staining, processes that require fixation or DNA intercalation and have variable efficacy in illuminating rDNA loci structure [17]. rDNA organization is potentially a useful visual biomarker of cellular state if we can grasp how the morphology reflects nucleolar activity.

We developed a quantitative imaging approach to investigate rDNA spatial organization in fission yeast. We selected fission yeast as our model due to the ease of genetic manipulation, its shared regulatory pathways with mammals, and the presence of two rDNA arrays as opposed to the single array in budding yeast. The two fission yeast rDNA arrays are found at the ends of chromosome 3, with a total of 80–120 rRNA encoding genes in WT lab strains. We utilized a DNA-binding protein capable of recognizing overtwisted DNA to quantify rDNA spatial organization over the cell cycle. Through the manipulation of cell signaling pathways and RNA polymerase I activity, we determined that the rDNA arrays visually contract and expand in response to cellular regulation of ribosome output. For example, TOR hyperactivation generates an rDNA expansion phenotype suggesting a novel readout for upregulation of ribosome biogenesis. Taking advantage of a genome-wide deletion collection available in fission yeast, we screened thousands of non-essential gene deletions for rDNA spatial organization phenotypes. Unexpectedly, several large ribosomal protein (RPL) gene deletions generated the rDNA expansion phenotype. Characterization revealed a surprising resistance to TOR pathway inhibition, suggesting activation of compensatory pathways for RPL gene insufficiency. We identified genes in putative compensatory cell signaling pathways, including a conserved MAPK pathway previously characterized for responding to extracellular stress. Our comprehensive genetic approaches in fission yeast utilizing GapR-GFP allowed us to identify factors that regulate rDNA spatial organization and revealed compensatory signaling pathways that respond to RPL gene insufficiency.

## Results

### GapR-GFP is a marker for rDNA spatial organization

We endeavored to generate a marker for quantification of rDNA spatial organization that would not disrupt rDNA function and could be used for live cell imaging. We turned to a bacterial DNA-binding protein called GapR. In its native organism *Caulobacter crescentus*, GapR binds to overtwisted DNA to recruit bacterial topoisomerases that relieve polymerase-induced torsional stress [18]. GapR recognizes this specific DNA topology through its structure, a dimer of dimers that forms a box around overtwisted DNA [18–20]. GapR has been used to identify overtwisted DNA, which has a smaller diameter than normal B-form DNA and occurs downstream of highly transcribed loci including the rRNA genes, in budding yeast [21].

Given the high rate of transcription of rRNA genes and the precedence in budding yeast, we predicted that GapR would bind downstream of rRNA genes in fission yeast. A GapR gene, codon optimized for *S. pombe*, was fused at the C terminus with GFP followed by a nuclear localization sequence (NLS) and integrated into the adenine locus under the control of a β-estradiol-inducible promoter [22]. GapR-GFP in wild-type (WT) fission yeast cells localized to the bulk chromatin as well as the rDNA arrays on both ends of chromosome 3 (Fig 1A). We

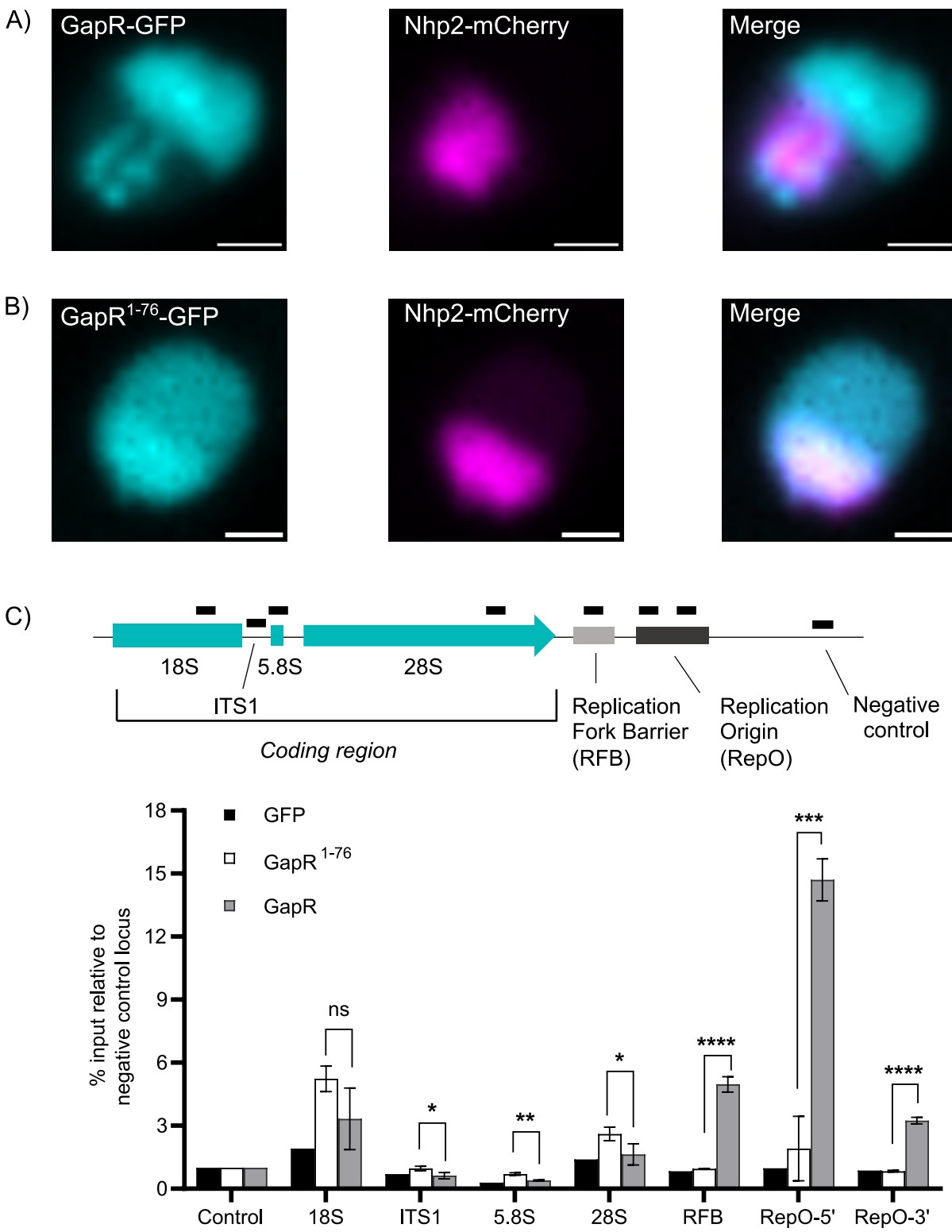

**Fig 1. GapR-GFP is a marker for rDNA spatial organization.** (A) A wild-type (WT) fission yeast nucleus expressing GapR-GFP or (B) GapR$^{1-76}$ and Nhp2-mCherry is shown. The scale bar represents 1μm. (C) ChIP-qPCR of GapR-GFP, GapR$^{1-76}$ control, and GFP-NLS enrichment at the rDNA locus is shown, with amplicon positions depicted by black bars. The bars show the mean %IP for the three replicates with error bars depicting standard deviation. Values above bars indicate p-value determined by unpaired t-test comparing GapR$^{1-76}$ and GapR, with ns = not significant, *p<0.05, **p<0.01, ***p<0.001, and ****p<0.0001.

marked the nucleolus with mCherry-tagged Nhp2, an established nucleolar marker in fission yeast [23]. Nhp2 functions as an rRNA-processing protein and is found in the nucleolar compartment throughout the entire cell cycle [24]. As a control for non-specific DNA binding, we expressed a truncated GapR-GFP protein (GapR$^{1-76}$) unable to recognize overtwisted DNA [19,21]. The truncated GapR$^{1-76}$ mutant is unable to highlight the two rDNA arrays but does localize to the nucleus and nucleolus, with some accumulation in the nucleolus, as often observed for exogenous proteins (Fig 1B). To characterize GapR-GFP as a marker for rDNA analysis, we examined GapR-GFP binding by ChIP-qPCR using the strains expressing either full-length or truncated GapR. GapR-GFP binds at the Replication Fork Barrier and Replication Origin, sequences that are downstream of the transcribed coding region and predicted to be overtwisted (Fig 1C). Furthermore, we observed a reduction in GapR-GFP enrichment in conditions where rDNA transcription is decreased (S1 Fig), indicating that GapR-GFP binds in a transcription-dependent manner in fission yeast.

The utility of GapR as a marker relies on a lack of interference with rDNA function. By binding downstream of the rRNA coding region, GapR-GFP is unlikely to interfere with RNA pol I activity. To examine the effect of GapR-GFP binding on rRNA transcription, we measured nascent and steady-state rRNA transcripts in GapR-GFP and GFP-NLS expressing strains. The latter served as a control for effects of the β-estradiol inducible expression system on rRNA transcription. Quantitative dot blot analysis of pulse-labelled RNAs showed that nascent transcript levels were unaffected by GapR-GFP expression (S2A and S2B Fig). qPCR analysis of rRNA transcripts confirmed that GapR-GFP expression does not impact steady-state rRNA transcript levels (S2C Fig). Altered rRNA production and ribosome biogenesis would be expected to negatively affect cell cycle progression and growth. Analysis of the distribution of cells in the cell cycle and growth in liquid culture demonstrates that GapR-GFP expression has minimal effects (S2D and S2E Fig). Analysis of nucleolar volume indicates that GapR-GFP expression does not impact nucleolar size (S2F Fig). These results indicate that GapR-GFP can mark rDNA arrays for imaging analysis without negatively impacting rDNA function or cell health.

## rDNA spatial organization is dynamic throughout the fission yeast cell cycle

Next we determined the utility of GapR-GFP as a live marker of rDNA organization over the fission yeast cell cycle. Timelapse imaging of an asynchronous culture of WT cells demonstrates GapR-GFP is associated with rDNA, as well as all nuclear DNA, at all cell cycle stages (Fig 2A). To assess rDNA and nucleolar morphology, we quantified four parameters. First, we quantified the volume of the rDNA arrays, defined as the GapR-GFP signal found within the mCherry marked nucleolus. Second, we determined the GFP signal intensity of the rDNA, defined as the average GapR-GFP signal intensity found within the nucleolus. To account for variability in cell size, a ratio of the rDNA volume to total nuclear volume was calculated. To account for variability in GapR-GFP protein levels, the rDNA mean GFP intensity was normalized to the total nuclear mean GFP intensity. Third, we examined the extension of the rDNA arrays, which we quantified as the distance between the center of the rDNA arrays and the center of the bulk chromatin outside the nucleolus. Finally, to examine nucleolar morphology, we measured the volume of the nucleolus using Nhp2-mCherry. To account for variability in cell size, we normalized Nhp2-mCherry volume to total nuclear volume using the total nuclear GapR-GFP signal. All of these parameters allowed us to characterize rDNA organization and nucleolar morphology over the cell cycle, as well as under different environmental and mutant conditions.

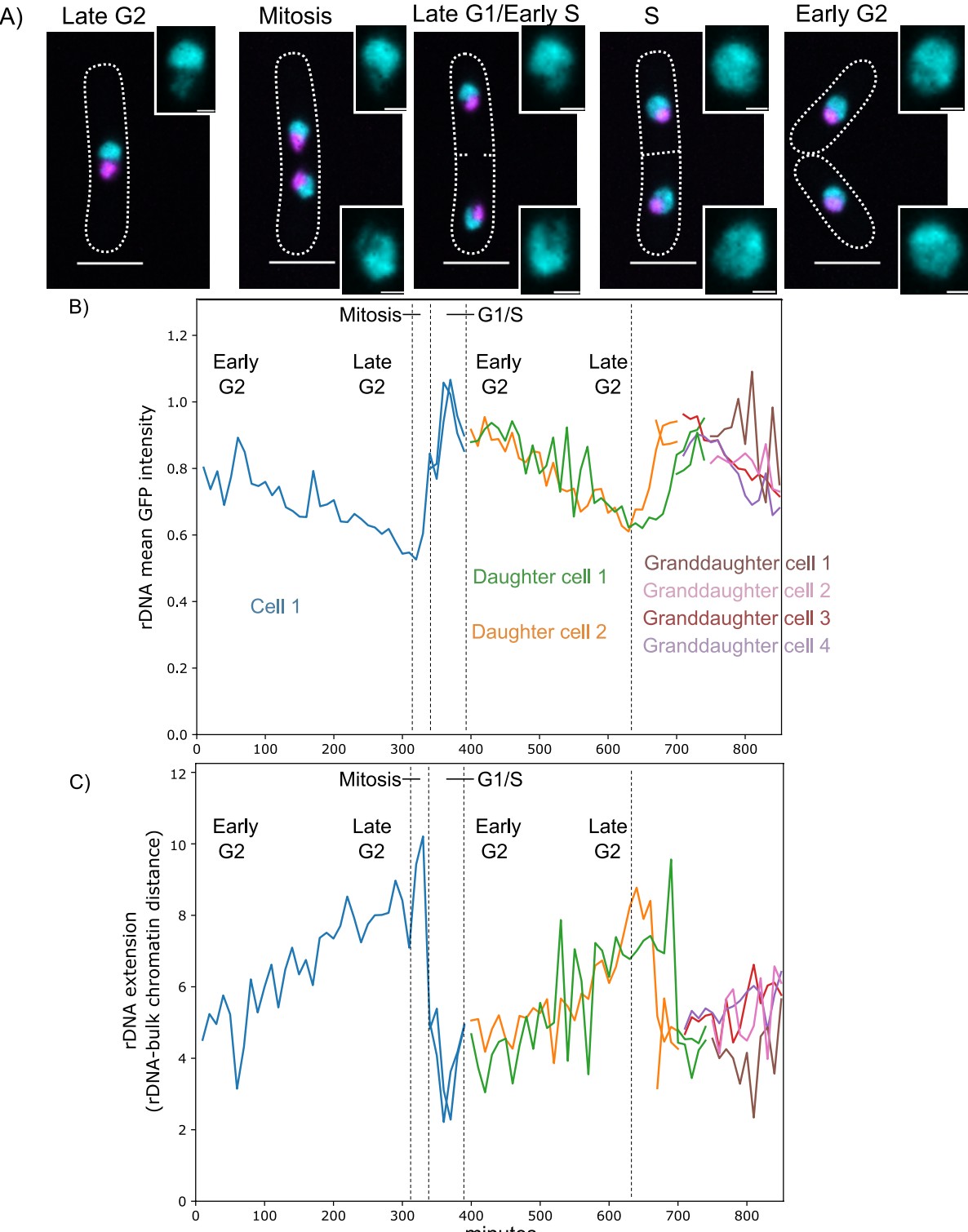

**Fig 2. rDNA spatial organization is dynamic across the fission yeast cell cycle.** (A) Timelapse images of WT yeast expressing GapR-GFP and Nhp2-mCherry representing each cell cycle stage are shown. Scale bars represent 6μm for the images with cell outlines and 1μm for the inset. (B-C) Plots show quantification of rDNA spatial organization over time. Each line represents a single nucleus while line color represents a single cell. The time between frames is 10 minutes. Key cell cycle transitions are marked by dashed lines for the first cell cycle. (B) Plot depicting quantification of the rDNA mean GFP intensity ratio over time. (C) Plot depicting quantification of rDNA extension (the distance between the midpoint of the rDNA and the midpoint of the bulk chromatin) over time.

Quantification of rDNA and nucleolar parameters described above has the potential to reveal dynamic organization of the rDNA arrays throughout the cell cycle. The intensity of nucleolar GapR-GFP appears to sharply increase during mitosis and is inversely correlated to rDNA extension (Fig 2B and 2C). These parameters likely reflect increased rDNA array condensation for chromosome segregation, with DNA condensation bringing GFP molecules in closer proximity to increase the average GFP intensity (Fig 2B). Following mitosis, GFP intensity peaks in late G1 or early S phase before declining. This decline in GFP intensity may represent chromosome de-condensation. Rapid GapR-GFP turnover on DNA would complicate this interpretation. FRAP analysis in *C. crescentus* revealed that GapR is stably bound to DNA until a replication fork is encountered [25]. We did not observe recovery following photobleaching of the nucleolar pool of GapR-GFP in *S. pombe* cells during G2, indicating a lack of turnover (S3 Fig). The stability of binding suggests GapR-GFP may be an imperfect reporter for rapid changes in transcriptional activity in a single cell cycle and we do not use GapR binding to directly infer overtwisted DNA or transcription rate in our study. However, stable binding is a convenient feature for a live marker for rDNA organization and is consistent with DNA-binding attributes of GapR in *C. crescentus*. Furthermore, GapR-GFP is advantageous as a live imaging marker because it is an inducible exogenous protein and does not require tagging an endogenous gene which could compromise functionality. Moreover, using GapR-GFP we have demonstrated that the rDNA spatial organization is dynamic throughout the fission yeast cell cycle.

## Nucleolar GapR-GFP reflects rDNA copy number

To further characterize GapR-GFP as a marker of the rDNA, we examined if its signal reflects copy number variation. In WT *S. pombe*, the rRNA genes exist in tandem arrays averaging 80–120 gene copies per cell. We isolated yeast colonies containing low (58), medium (80), or high (147) 28S gene copy number as determined by digital droplet PCR [26]. To control for the variation in rDNA morphology during the cell cycle, we limited our analysis to cells in G2, which we separated into early and late G2 based on cell length (see methods). Our results indicate that GapR-GFP volume increases with rDNA copy number, indicating that GapR-GFP volume reflects rDNA volume (Figs 3A and S4A and S4D). Intriguingly, we observe a similar increase in mean GapR-GFP intensity with copy number (Figs 3B and S4B and S4E). We speculate that additional nontranscribed repeats contribute to condensation and increased GapR-GFP signal intensity. In contrast, 28S gene copy number does not increase nucleolar volume as measured by Nhp2-mCherry, indicating additional repeats are not expanding nucleolar size (Figs 3C and S4C and S4F). Altogether, we present a novel marker for rDNA spatial organization in fission yeast that is distinct from nucleolar volume and can be used for live cell imaging without disturbing rDNA function.

## rDNA spatial organization can reflect ribosome biogenesis activity

We hypothesized that rDNA spatial organization reflects not only 28S copy number, but may report on ribosome biogenesis activity. To this end, we manipulated ribosome biogenesis activity and examined spatial organization of the arrays. First we examined rDNA morphology under glucose starvation. Glucose-starvation activates a stress response that reduces ribosome biogenesis, particularly transcriptional repression of rRNA and ribosomal protein mRNAs [27,28]. Analysis of total nascent transcript levels indicated transcriptional shut-off with 2 hours of glucose starvation (S5 Fig), consistent with a previous study [29]. We observed rDNA compaction in response to glucose starvation. The rDNA arrays condensed, as measured by a decrease in rDNA volume and increase in mean GFP intensity, and re-localized to the

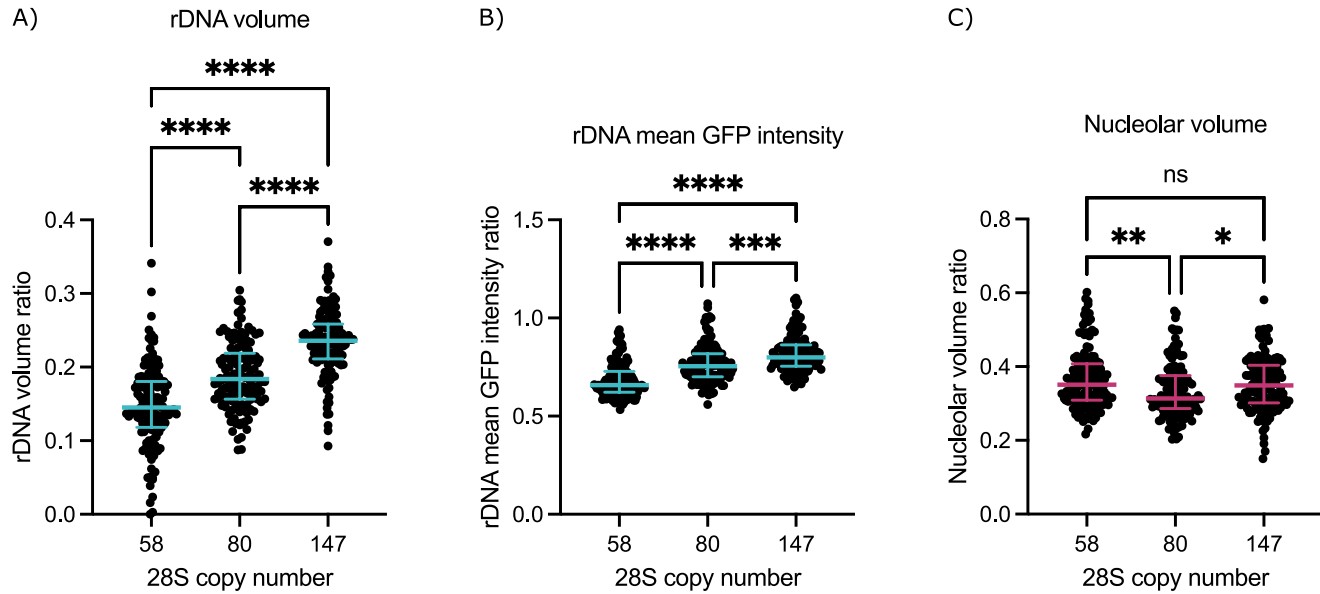

**Fig 3. rDNA spatial organization reflects 28S gene copy number.** (A-C) Plots show quantification of late G2 cells in strains with low (58), medium (80), or high (147) 28S gene copy number. Plots show the rDNA volume ratio (A), rDNA mean GFP intensity ratio (B), and mCherry volume ratio (C) for a representative biological replicate (n>135 cells). For this figure and all subsequent imaging quantification, the rDNA and mCherry volumes (pixels$^3$) were normalized to the total nuclear GFP volume (pixels$^3$). Similarly, the rDNA mean GFP intensity value (arbitrary units) was normalized to the total nuclear mean GFP intensity (arbitrary units). Bars represent the median and interquartile range. Statistical significance of the 3 biological replicates was determined by Kruskal-Wallis test (one-way ANOVA): ns = not significant, *p<0.05, **p<0.01, ***p<0.001, ****p<0.0001.

periphery of the nucleolar compartment, especially in early G2 cells (Figs 4A and 4C and S6). Interestingly, the extent of condensation varied by cell cycle stage (early versus late G2), implying there may be cell cycle specific responses of the rDNA to glucose starvation (Figs 4A and S6). The nucleolar compartment also showed decreased volume with glucose starvation (Figs 4D and S6).

Nucleolar shrinking and rDNA condensation are both conserved responses to the inhibition of RNA pol I activity previously described in budding yeast and humans [16,30,31]. Next, to further examine how the rDNA responds specifically to transcriptional repression in fission yeast, we examined rDNA organization following treatment with an RNA polymerase I inhibitor, Actinomycin D. Condensation and peripheral re-localization of the rDNA was observed with Actinomycin D treatment (S7A–S7C Fig). Like glucose starvation, Actinomycin D treatment also caused nucleolar shrinkage (S7D Fig). Additionally, Actinomycin D treatment causes an increase in Nhp2 signal in the nucleoplasm, although most Nhp2 remains nucleolar (S7A Fig). These results indicate that in the absence of transcription, the rDNA arrays reorganize to condense and relocate to the periphery of the shrinking nucleolar compartment. These inactive rDNA foci, as well as the diffusion of nucleolar proteins into the nucleus, are reminiscent of mammalian stress caps, in which the rDNA loci condense and re-localize to the nucleolar periphery upon RNA polymerase I inhibition, suggesting a common evolutionarily conserved organizational response to transcriptional inhibition [32,33].

To determine if the rDNA condensation phenotype we observed with glucose starvation is reproducible with an alternative endogenous protein, we fluorescently tagged an exogenous copy of the RNA pol I subunit Rpa43 in a WT strain. We found that with glucose starvation, Rpa43-BFP also condenses, confirming the re-organization with a second marker. However, the nucleolar Rpa43 signal is more diffuse relative to the rDNA arrays, suggesting some Rpa43

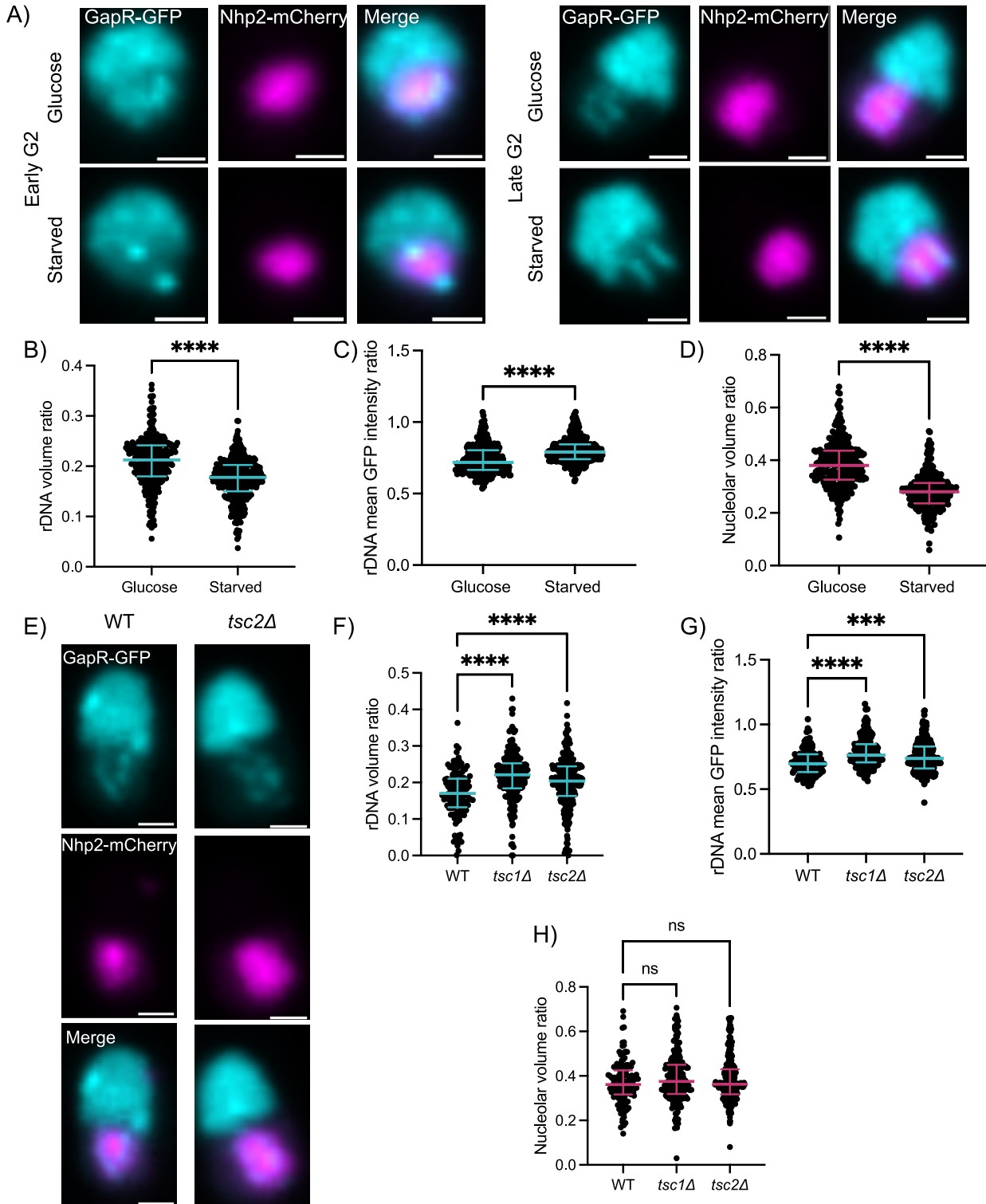

**Fig 4. rDNA spatial organization reflects nutritional status.** (A) Images of WT nuclei expressing GapR-GFP and Nhp2-mCherry with or without 1.5 hour of glucose starvation. The scale bar represents 1μm. (B-D) Plots show quantification of conditions in 2A. Plots show the individual values with bars representing the median and interquartile range for rDNA volume ratio (B), rDNA mean GFP intensity ratio (C), and mCherry volume ratio (D) for a representative biological replicate (n>314 cells). (E) Representative images of WT and mutant nuclei in G2 expressing GapR-GFP and Nhp2-mCherry are shown. The scale bar represents 1μm. (F-H) Plots show the rDNA volume ratio (F), rDNA mean GFP intensity ratio (G), and

mCherry volume ratio (H) for a representative biological replicate (n>127 cells). Statistical significance of the 3 biological replicates was determined by Kruskal-Wallis test (one-way ANOVA): ns = not significant, *p<0.05, **p<0.01, ***p<0.001,****p<0.0001.

is not bound to the rDNA (S8A and S8B Fig). The binding stability and position of GapR outside of rRNA genes makes it an exceptional marker for assessing rDNA organization in a variety of stressors.

To examine rDNA organization in the context of increased ribosome biogenesis, we manipulated the cell signaling pathways that regulate RNA polymerase I activity. The TOR pathway is a key nutrient sensing cell signaling pathway that drives ribosome biogenesis including transcription of rRNA genes through RNA pol I recruitment [34,35]. Hyperactivation of the TOR pathway can be achieved by knocking out either *tsc1* or *tsc2*, two negative regulators of the TORC1 complex responsible for driving cell proliferation [36]. We examined rDNA morphology in the absence of *tsc1* or *tsc2*, comparing strains that had approximately equal 28S gene copy number. We observed increased volume of the rDNA and mean GapR-GFP intensity; nucleolar volume was unaffected (Figs 4E–4H and S9). These results indicate increased ribosome biogenesis corresponds with expansion of the rDNA arrays independent of nucleolar volume. This expansion may be attributed to additional transcriptional machinery at active genes and/or an increased number of active genes.

## Genome-wide screen identifies large ribosomal proteins as regulators of rDNA spatial organization

Our results support a model in which rDNA morphology reflects both rRNA gene copy number and ribosome biogenesis activity. To date, no comprehensive genetic screen for regulators of rDNA spatial organization has been conducted, but GapR-GFP enables this search. We sought to identify novel genetic regulators of rDNA spatial organization through an imaging screen of the fission yeast Bioneer deletion collection [37]. The haploid Bioneer collection contains 3,400 strains with non-essential gene deletions. As proof of principle, we performed a pilot screen of mutants that we suspected could have rDNA organization phenotypes. We included 3 mutants for which we predicted altered ribosome biogenesis could be affected by loss of 1) RNA polymerase I transcription factor *reb1*, 2) ribosome assembly factor *mrt4*, and 3) large ribosomal protein gene *rpl3602*. Additionally, we included a strain with high rDNA copy number (*rtt109Δ*). Except for the *rtt109Δ* mutant, which had ~140 28S gene copies, we isolated single colonies for all mutants with comparable 28S copy number to WT. In all four strains we observed significantly altered rDNA volume and mean GFP intensity measurements compared to WT, demonstrating that deletion of ribosome biogenesis factors can significantly alter rDNA spatial organization (S10 Fig). We proceeded with screening the entire Bioneer deletion collection, using these four strains as positive controls for altered rDNA morphology. We mated a GapR-GFP Nhp2-mCherry haploid strain to the Bioneer collection in duplicate, sporulated the diploids, and selected haploid progeny that inherited GapR and Nhp2 markers and a gene deletion (Fig 5A). Strains were imaged live to preserve the structural integrity of the nucleus in the absence of fixative agents.

To call hits, we set normal thresholds (+0.33 or -0.25 standard deviation) for rDNA parameters by averaging the median rDNA volume and rDNA mean GFP intensity values for 8 WT controls (the same WT strain imaged on 8 separate plates) (white dots, center Fig 5B). The median rDNA volume and rDNA mean GFP intensity values were averaged for each deletion mutant (independently derived and imaged in duplicate with a minimum of 30 cells) (Fig 5B). We quantified rDNA phenotypes for 2,976 strains. 123 strains had rDNA volume and mean

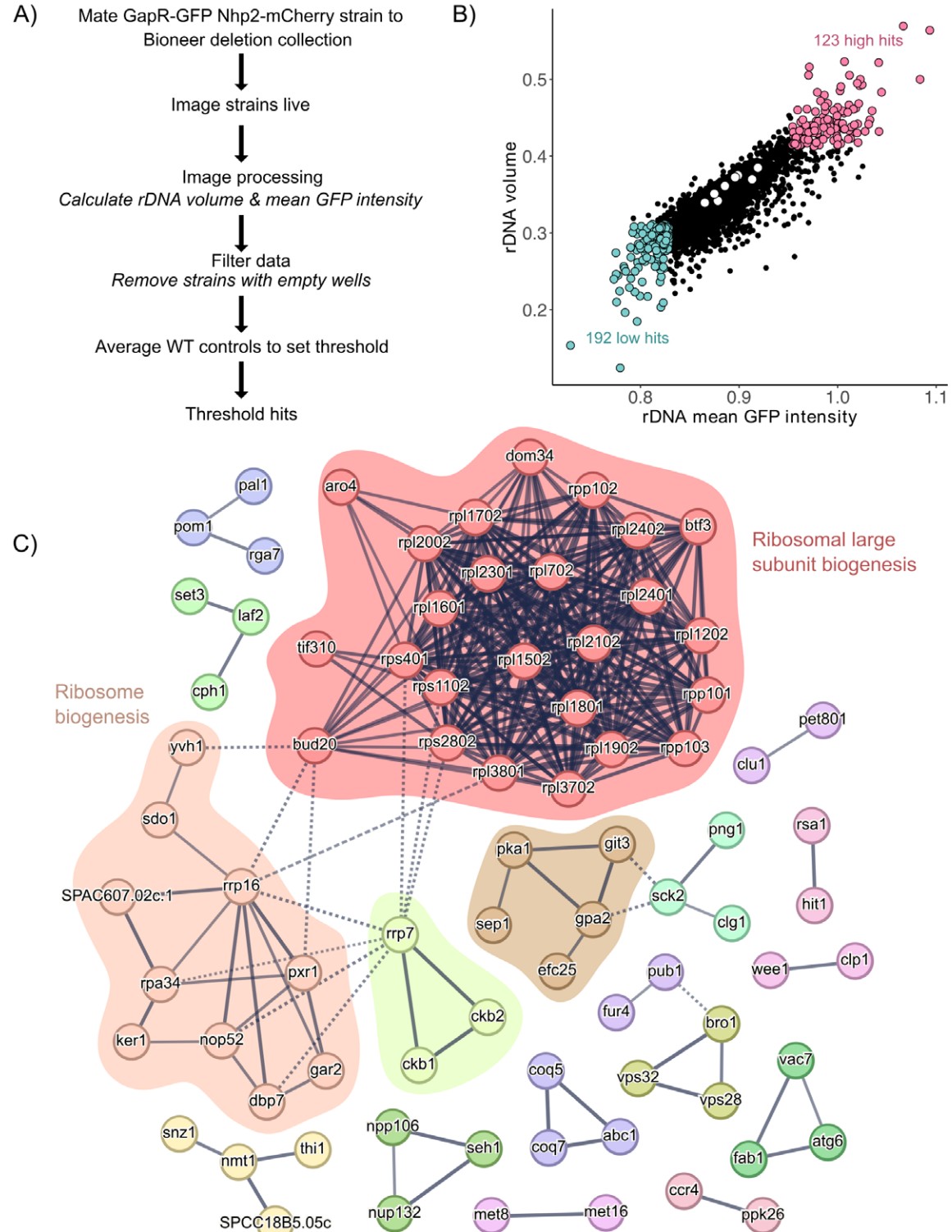

**Fig 5. Genome-wide imaging screen for regulators of rDNA spatial organization.** (A) An outline of the approach used to generate the GapR-GFP mutant library and subsequent screening is shown. (B) Screen hits were identified by setting a threshold above and below an average of 8 WT (white circles) values. Each point represents the average rDNA quantification for 2 mated replicates of each mutant. The plot was generated in R version 4.1.2. (C) STRING Network analysis of the genes deleted in 123 "high" rDNA phenotype hits is shown. The line color indicates confidence (strength of data support) using a high minimum required interaction score (0.7). Node color was

determined by MCL clustering (inflation parameter = 2). Highlighted clusters include RPL ribosomal assembly (25 nodes), ribosome biogenesis (10 nodes), cAMP-mediated signaling (5 nodes), and regulation of transcription by RNA pol I (3 nodes). Unconnected nodes are not shown.

GFP intensity measurements above the WT threshold (Fig 5B and S1 Table). 192 strains had low rDNA and low mean GFP intensity values (Fig 5B and S1 Table). STRING Network Analysis of the genes deleted in strains with increased rDNA volume and mean GFP intensity–called "high" hits–identified several related clusters. The largest cluster included genes encoding large ribosomal proteins, hereafter RPL proteins, and was linked to a smaller cluster of ribosome biogenesis factors (Fig 5C). Some known regulators of rDNA activity, e.g. *hmo1*, were not identified in our screen. Variability in rDNA gene copy number, incorrect or missing deletions in the library, and a stringent threshold for rDNA phenotypes are a few reasons why some factors were not identified as screen hits. Additionally, we were unable to analyze some strains due to insufficient cell number (<30 cells) or because they did not survive generation of the screening library; these strains are listed in S2 Table. Our findings suggest rDNA features can be used as a biomarker to detect altered ribosome biogenesis.

Gene ontology of the genes deleted in the 192 hits below WT thresholds failed to identify significantly enriched pathways. Given the variability in the inheritance of rDNA copy number following meiosis, we suspect that some of these hits are the result of progeny inheriting decreased rRNA gene copy number. We focused our validation efforts on the 123 "high" hits, with phenotypes that suggest altered ribosome biogenesis activity.

To validate our results, we remated the 123 strains that yielded the "high" phenotype to a prototrophic GapR-GFP query strain and performed the same imaging analysis. Given our interest in the RPL mutant phenotype, we expanded our validation screen to include all non-essential RPL and small ribosomal protein (RPS) mutants, some of which were not recovered in the initial GapR-GFP screening library (S3 Table). For the validation screen, we included 3 WT strains with variable rDNA copy number to generate an average "WT" rDNA phenotype threshold (S3 Table, strains 738, 740, 746 with 28S gene copies ~62, ~105, and ~140, respectively). Accounting for variability in rRNA gene copy number is critical, as copy number variation occurs following transformation or meiosis in budding yeast [38,39]. Using this averaged threshold, we validated the high rDNA mean GFP intensity and rDNA volume phenotypes in several mutants involved in ribosome biogenesis, with many other strains validating for just high GFP intensity (S11 Fig and S3 Table). This included nine RPL deletion mutants. The finding that RPL mutants presented with a phenotype similar to the *tsc1* deletion which hyperactivates ribosome biogenesis was a conundrum, since these deletions might be expected to curtail ribosome biogenesis.

Given that increased rRNA gene copy number is associated with increased rDNA volume and mean GFP intensity, we further quantified rDNA phenotypes in strains screened for equivalent 28S copy number. We selected six RPL mutants with increased rDNA volume and mean GFP intensity in both the initial and validation screens for further analysis. Two RPL mutants and two RPS mutants, that were not identified as hits, were included as negative controls. rDNA volume and mean GFP intensity, as well as nucleolar volume, were consistent with initial screen results (Figs 6 and S12), suggesting the phenotype is not due to copy number. The increase in nucleolar volume contrasts with the *tsc1Δ* phenotype, which did not show increased volume, suggesting RPL deletions generate a unique phenotype. Moreover, we noted that phenotypic strength is a reproducible feature of a gene deletion e.g. *rpl3801Δ* consistently has a stronger phenotype than *rpl2102Δ*.

To determine if the rDNA expansion phenotype in RPL mutants is reproducible with a second marker, we fluorescently tagged the RNA pol I subunit Rpa43 in WT and *rpl3801* deletion

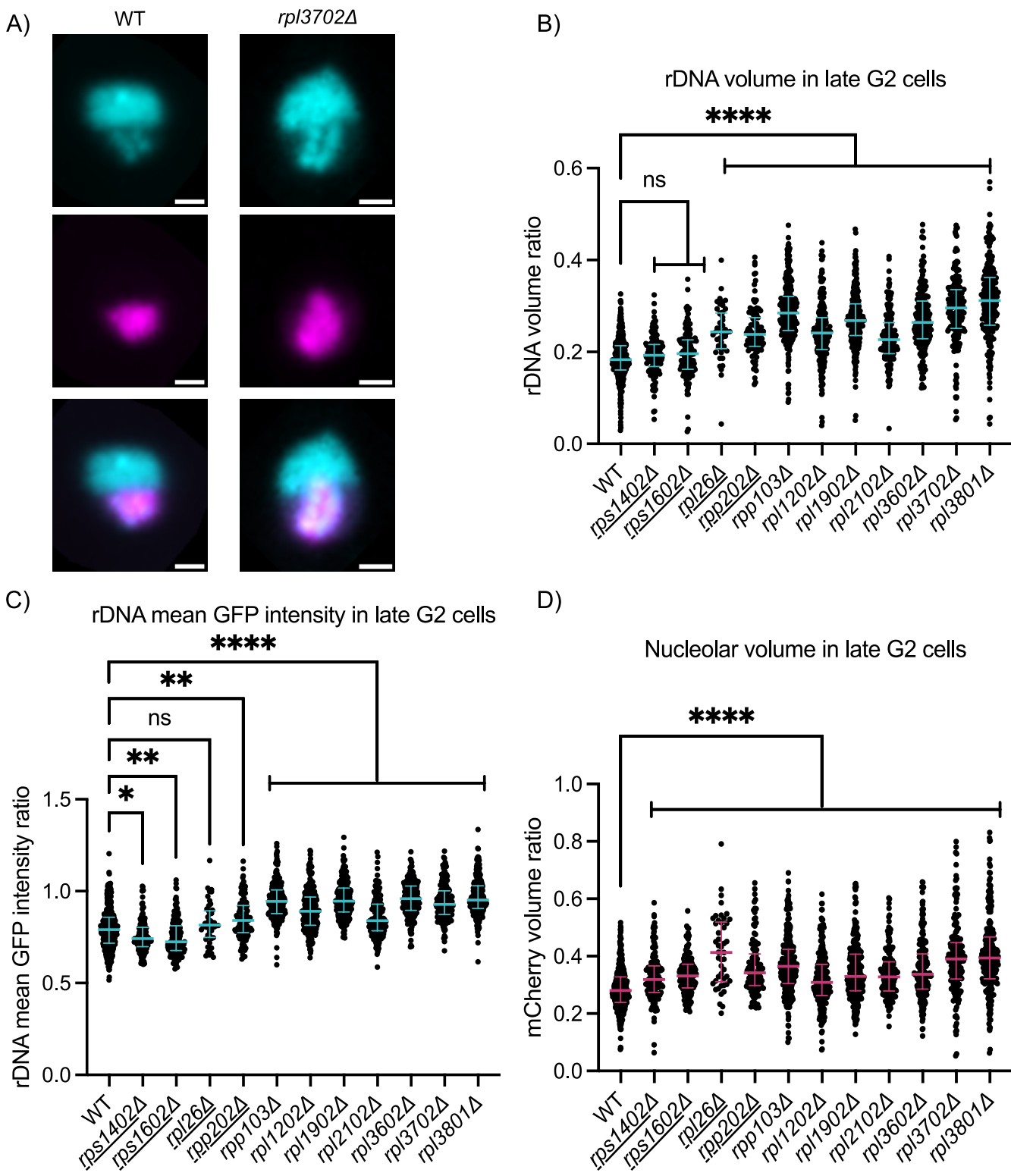

**Fig 6. Validation of rDNA spatial organization phenotype in RPL gene deletion strains.** (A) Images of GapR-GFP and Nhp2-mCherry nuclei in WT and *rpl3702Δ* strains. The scale bar represents 1μm. (B-D) Plots show quantification of ribosomal protein deletion strains identified from the initial and validation screens. RPS and RPL control strains, defined as RP mutants not identified in the GapR imaging screen, are underlined. The rDNA volume ratio (B), rDNA mean GFP intensity ratio (C), and mCherry volume ratio (D) for a representative biological replicate is shown (n>50 cells). Bars represent the median and interquartile range. Statistical significance was determined by Kruskal-Wallis (one-way ANOVA): ns = not significant, *p<0.05, **p<0.01, ***p<0.001, ****p<0.0001.

strains. We were able to reproduce the increased rDNA volume phenotype using Rpa43-BFP (S13 Fig). Taken all together, these data suggest RPL gene deletion induces a robust visual phenotype at the rDNA.

We examined whether RPL gene deletion impacts rDNA transcriptional activity by measuring nascent transcript levels in *rpl3801Δ*, the mutant with the strongest visual phenotype (Fig 6B and 6C). There was no significant difference in nascent transcript levels between *rpl3801Δ* and WT (S14 Fig). We further tested whether altered transcriptional activity was suggested by changes in DNA topology using GapR-GFP binding. We observed weak but significantly increased GapR-GFP binding at one of three rDNA amplicons in one RPL gene deletion mutant (*rpl3602Δ*) and one ribosome assembly factor deletion mutant (*mrt4Δ*) that exhibited the high rDNA volume and mean GFP intensity phenotype, (S15 Fig). Overall these results do not support large changes in transcriptional activity at the rDNA in the mutant backgrounds. Thus, we interpret the changes in rDNA organization simply as dysregulated ribosome biogenesis. Future studies will be required to elucidate specific molecular defects in ribosome biogenesis.

Ribosomal proteins are essential for cell viability. In fission yeast, many ribosomal protein genes are present as paralogs with 1–3 copies per gene. This redundancy allows for ribosomal protein gene deletion, although levels are likely affected. Studies in budding and fission yeast determined that the expression of each paralog is not necessarily equal, and the predominantly expressed paralog may depend on cell cycle state or exposure to stress [40,41]. RNA-seq analysis of a WT asynchronous culture highlights all ribosomal protein gene paralogs are expressed at some level, with *rpl1603* being an exception (S4 Table). Thus, ribosomal protein gene deletion should be considered an insufficiency phenotype with the level of insufficiency dependent on which paralog was deleted.

### RPL gene deletion alters signaling pathways including the cell integrity MAPK pathway

Our screen demonstrated that deletion of RPL genes created a rDNA phenotype similar to but distinct from the TOR activation phenotype (Fig 4E–4H), suggesting ribosome biogenesis could be in overdrive. We hypothesized that RPL gene insufficiency could trigger a compensatory response in ribosome biogenesis activity. Given the well-established role of the TOR pathway as a driver of ribosome biogenesis, we tested whether TOR pathway activity was altered in RPL mutants. We examined RPL mutant growth in the presence of Torin1, a highly specific ATP-competitive inhibitor of both the TORC1 and TORC2 complexes [42,43]. We observed Torin1 resistance in several RPL mutants identified in our screen relative to WT and RPS mutants (Fig 7A), suggesting overactivation of regulatory pathways allowing for survival upon TORC inhibition. Interestingly, RPL mutants with the strongest Torin1 resistance also appeared to have the highest rDNA volume and mean GFP intensity, e.g. *rpl3801Δ* (Figs 6B and 6C and 7A). The variation in both rDNA and Torin1 resistance phenotypes may become clearer as the canonical and moonlighting functions of ribosomal proteins, as well as expression of individual paralogs, is better characterized in fission yeast. Given the number of pathways that regulate ribosome biogenesis, we suspected that multiple signaling pathways could be altered in response to RPL gene deletion.

To identify which cell signaling pathways are altered in RPL mutants, we performed a Synthetic Genetic Array experiment designed to identify factors essential for Torin1 resistance. We mated 5 RPL mutants with Torin1 resistance to 110 non-essential gene deletions in candidate signaling pathways. One RPL and two RPS mutants lacking Torin1 resistance were also included as controls, giving us a total of 8 query strains. Candidate pathways included the

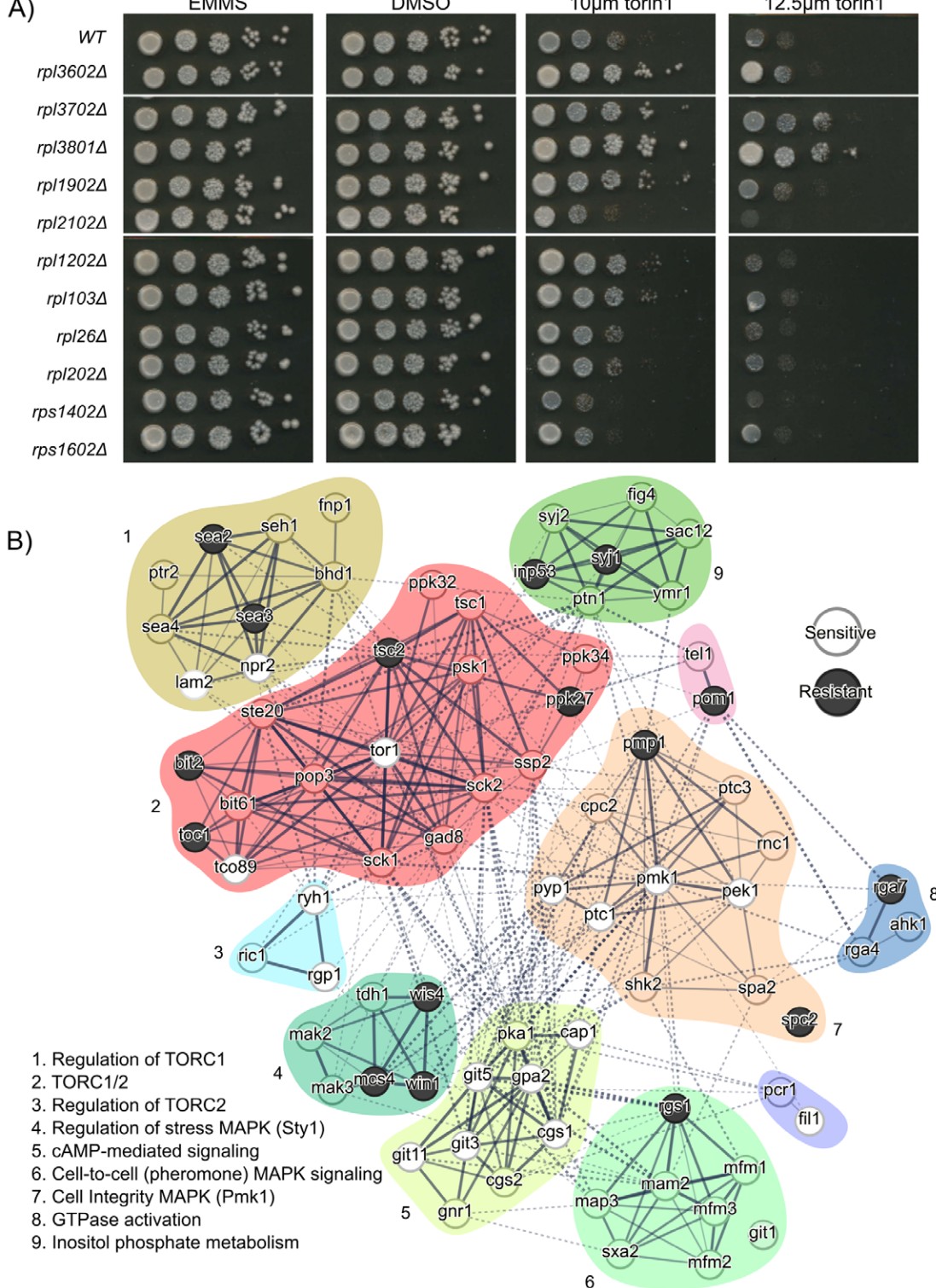

**Fig 7. RPL gene deletion screen hits alter ribosome biogenesis signaling pathways.** (A) WT and ribosomal protein deletion strains were spotted onto an EMMS plate without drug, with DMSO, or with 10μm or 12.5μm Torin1. Resistance to Torin1 was defined as growth greater than the WT strain for each condition (i.e. for 10μm Torin1, growth in columns 4 or 5 indicated resistance). (B) STRING Network Analysis depicting non-essential ribosome biogenesis signaling factors screened for modulating resistance to Torin1 in ribosome protein mutants. Signaling factor mutants that are essential for Torin1 resistance are highlighted in white, while factors that repress Torin1 resistance are highlighted in black.

cAMP/PKA pathway, MAPK stress pathways, PI3K pathway, and TOR pathway. The cAMP/PKA pathway works in parallel with the TOR pathway to drive ribosome biogenesis in response to nutrients [44]. The three MAPK pathways respond to various environmental stressors; some of these pathways require the PKA pathway for activation [45,46]. We scored growth by measuring colony size of each double mutant on medium with Torin1 or medium with DMSO as a control. We calculated the fold change of growth on these two mediums to identify strains that were sensitive or resistant to Torin1 (S16 Fig, see materials and methods and S5 Table). From this experiment we generated two outputs: 1) factors that were essential for Torin1 resistance in RPL mutants, and 2) factors that inhibited Torin1 resistance in RPL and RPS mutant controls (Fig 6B and S5 Table). The latter included several factors known to inhibit TOR, such as Toc1 and Tsc2 (Fig 7B, cluster 2), validating our approach and reaffirming that overactivation of the TOR pathway can drive Torin1 resistance [43].

Our genetic screen indicated all candidate signaling pathways contain at least one factor that can provide input into Torin1 resistance. One pathway highlighted by synthetic lethality for RPL gene deletion and Torin1 resistance is the Pmk1 stress activated MAPK pathway (cluster 7, Fig 7B). Pmk1 is the MAP kinase for the Cell Integrity Pathway, orthologous to the ERK1/2 pathway in mammalian cells, a pathway that upregulates ribosome biogenesis and stimulates mTOR signaling when activated [47]. *pmk1* deletion abolished Torin1 resistance in RPL mutants (Fig 7B), suggesting a Pmk1-dependent pathway can support ribosome biogenesis in RPL mutants when TOR is inhibited. Consistent with this idea, factors known to activate Pmk1, including *pek1*, *ryh1*, and several cAMP/PKA components, also abolished Torin1 resistance when deleted in RPL mutants (cluster 5, Fig 7B) [45,48,49]. Additional genetic interactions also support the involvement of this pathway in Torin1 resistance. *Rga7*, *pmp1*, and *pom1* negatively regulate Pmk1 and generated Torin1 resistance when deleted in Torin1 sensitive strains, implicating activation of Pmk1 in Torin1 resistance (Fig 7B) [50,51]. To determine how Pmk1 pathway components influence rDNA spatial organization, we revisited our screen results. *Rga7* and *pom1* deletion mutants, expected to activate Pmk1, were identified as having the "high" ribosome biogenesis phenotype observed in RPL mutants (Fig 5C). Our SGA results combined with our imaging screen confirm that many pathways provide regulatory cues for rDNA organization. Excitingly, we identified the Cell Integrity Pathway, and Pmk1 in particular, as a potential new positive regulator of ribosome biogenesis in fission yeast, although more work will be required to confirm this genetic inference.

## Discussion

In this work we developed a new tool capable of high-throughput imaging analysis of the rDNA arrays in live fission yeast cells. We demonstrate that spatial organization of the rDNA arrays reflects rRNA gene copy number, cell cycle phase, and likely cellular regulation of ribosome biogenesis activity, including rDNA condensation following RNA pol I inhibition and rDNA expansion upon TOR hyperactivation (Fig 8). Changes in rDNA organization during glucose starvation and in RPL deletion mutants were confirmed by two independent visual rDNA markers, GapR and Rpa43, lending confidence to these observations. Importantly, nucleolar volume was unaffected by perturbations like copy number and TOR activation, highlighting the usefulness and sensitivity of an rDNA marker as an imaging tool for nucleolar processes. Our broad investigation of rDNA spatial organization uncovered novel regulators of rDNA morphology including RPL genes. While some RPS genes were also hits, the preponderance of RPL genes is consistent with other studies suggesting the cellular response to RPS and RPL insufficiency can be distinct. In budding yeast, RPL insufficiency was found to upregulate the expression of protein degradation genes [52]. In human cell culture, a screen of

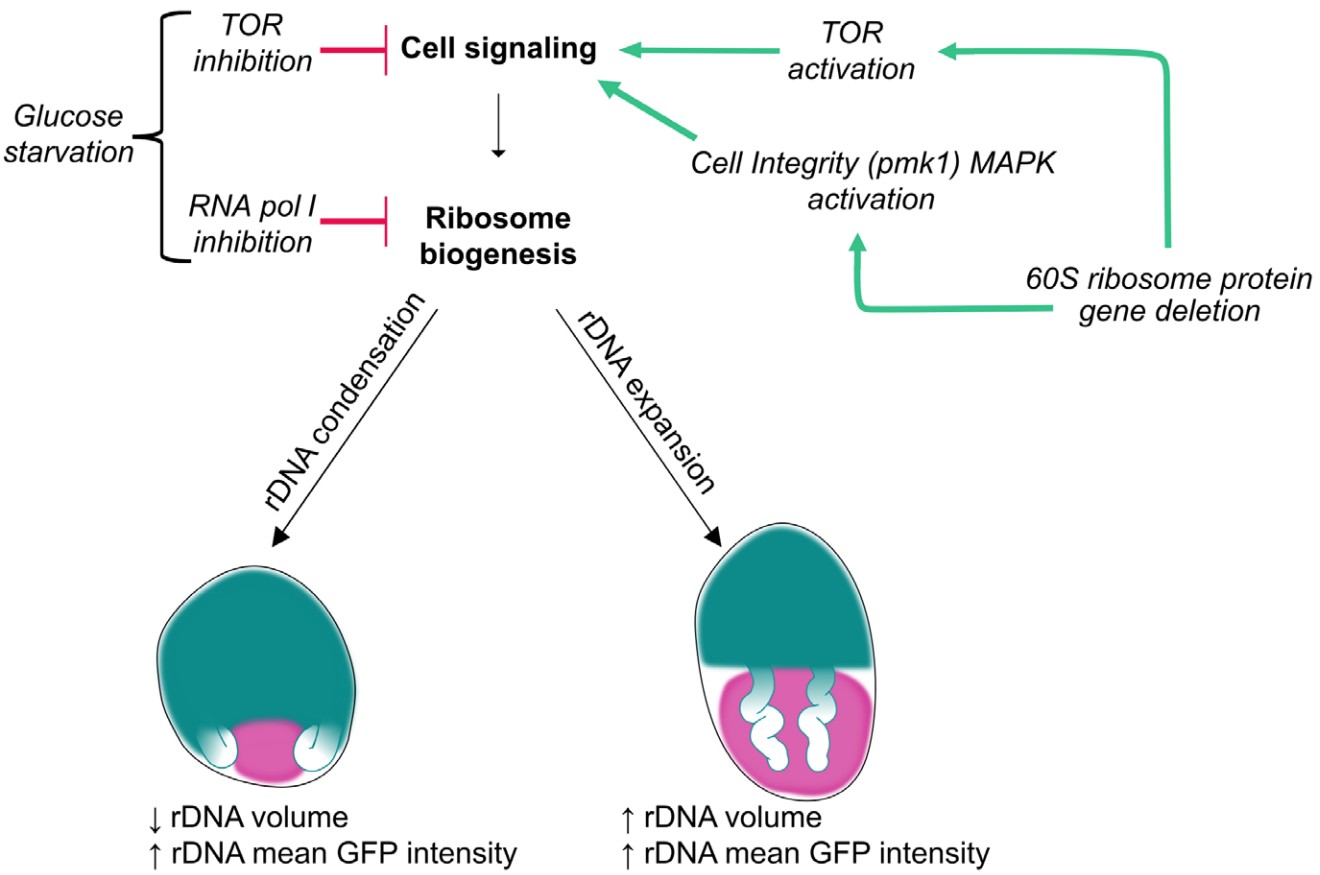

**Fig 8. Connections between regulatory pathways and rDNA morphology.** Decreased rDNA activity (red bars) through nutrient starvation or RNA pol I inhibition results in rDNA array condensation (GapR signal in teal, nucleolus in pink). TOR pathway activation or RPL gene deletion (green arrows) results in rDNA array expansion (GapR signal in teal, nucleolus in pink). The mechanism by which RPL gene deletion affects rDNA organization involves activation of the TOR and Cell Integrity MAPK pathways.

ribosomal protein knockdowns found RPL insufficiency had the greatest effect on nucleolar structure [53]. Investigation of the mechanism by which RPL gene deletion alters rDNA morphology suggests the insufficiency triggers a compensatory response in ribosome biogenesis mediated by many signaling pathways, including hyperactivation of a conserved MAPK pathway. Our study provides a comprehensive genetic and cell biological evaluation of regulators of rDNA organization, exposing many new players. These findings have important implications for human diseases associated with the loss of ribosomal protein subunits, particularly ribosomopathies and cancer.

The use of GapR-GFP in fission yeast provides both a novel imaging tool and model system to study fundamental aspects of ribosome biogenesis and chromosome organization. The ability to accurately assess rDNA morphology, particularly in high-throughput, is challenging. We attempted to generate several alternative markers to visualize the rDNA arrays but found that manipulation of most rDNA associated factors is detrimental to cell health, with Rpa43-BFP as the only successful alternate. The GapR-GFP inducible expression system has many advantages, including no significant functional costs, and robust fluorescence for live-cell imaging, avoiding the potential pitfalls of fixation while allowing the study of dynamic changes in chromosome organization. The GapR-GFP system can be quickly introduced into thousands of

yeast strains through plasmid integration or mating, a feature we used to our advantage to conduct a high-throughput imaging screen. The GapR-GFP tool reveals the spatial organization of the arrays, an organization that may reflect ribosome biogenesis activity, but this will require further characterization. In the future GapR-GFP could be adapted as a live marker of rDNA and other highly transcribed genes in other contexts, including in human cells.

Ribosome biogenesis is an essential and energy consuming cellular process necessitating multiple regulatory inputs. We identified and investigated a compensatory response to RPL gene deletion that includes resistance to TOR inhibition by Torin1. Our synthetic genetic screen implicates many signaling pathways in the compensatory response, including previously published regulatory factors Toc1 and Tsc2 [43]. We identified the Cell Integrity Pmk1 MAPK pathway as a novel modulator of Torin1 resistance. Known activators of the Pmk1 pathway include extracellular triggers like oxidative stress, osmotic stress, and cell wall damage [54]. RP gene deletion represents the first documented intracellular stress capable of activating the Pmk1 pathway. The orthologous mammalian pathway to the *S. pombe* Pmk1 pathway is a MAPK pathway controlled by ERK1/2, also known to respond to extracellular signals and upregulate ribosome biogenesis activity. How RP gene deletion alters the activity of the Pmk1 signaling pathway is unclear and deserves further investigation.

To our knowledge, we present the first example of RPL gene insufficiency causing a visual phenotype at the rDNA, consistent with a compensatory response in global cellular regulation of ribosome biogenesis. This finding has potential implications for human health. Insufficiency for ribosomal proteins occurs in several human disease contexts. This includes ribosomopathies, genetic disorders of dysregulated ribosome biogenesis that include mutations in ribosomal protein genes, and cancer. We speculate that ribosomal protein insufficiency may induce compensatory changes to allow for proliferation that can lead to oncogenesis. For example, individuals with ribosomopathies have bone marrow failure as well as a higher predisposition for blood cancer later in life, suggesting a transition from a hypoproliferative state to hyperproliferation [55]. RP genes are hemizygously lost in nearly half (43%) of cancers, [56], suggesting compensatory changes may create a selective advantage. Studies in zebrafish also identified elevated cancer incidence with RP haploinsufficiency, with the authors suggesting that RP genes may act as tumor suppressors [57]. Cancer cells are known for nucleolar hypertrophy, increased ribosome production, and altered cell signaling to boost RNA pol I activity [3]. Our results lead us to speculate that the compensatory response to RP gene insufficiency we observe in fission yeast may be conserved and contribute to oncogenic disease processes in higher eukaryotes.

## Materials and methods

### Media preparation and yeast culture

All strains used are listed in S6 Table. Glycerol stocks were initially streaked onto Yeast Extract supplemented with adenine, histidine, leucine, lysine, and uracil. All experiments were conducted using Edinburgh Minimal Media supplemented (EMMS) with adenine, leucine, lysine, uracil, and histidine with 2% glucose, potassium hydrogen phthalate, and ammonium chloride unless otherwise indicated. Dilution assays to assess growth were conducted using fivefold serial dilutions starting at an OD of 0.2 spotted onto EMMS plates. Strains were grown at 30˚C. For glucose starvation assays, cells were pelleted at 2000g for 3-5min, washed with EMMS-No glucose, then resuspended in EMMS-No glucose for the desired time (generally 1.5–2 hours).

## Strain generation

Standard lithium acetate transformation was used for plasmid or PCR fragment integration. The GapR-GFP query strain was generated by plasmid integration. The *C. crescentus* GapR coding sequence was ordered as an *S. pombe* codon-optimized g-block from IDT. The g-block was recombined into the pFS462 (Addgene) plasmid using Gibson assembly. The pFS462 plasmid was integrated following AflII digestion. The pFS461 plasmid (Addgene) was integrated at the *leu1* locus following XhoI digestion to allow for inducibly controlled GapR-GFP expression using the β-estradiol system [22]. For strain 646, the pFS462 *his7* gene was replaced with *ade6* and integrated following AatII digestion. The GapR$^{1-76}$ deletion plasmid was generated by site-directed mutagenesis using plasmid pFS462-GapR as the template (S7 Table). Nucleolar marker Nhp2 was tagged at the endogenous locus with a C-terminal mCherry tag, selected by clonNatR.

Strains with gene knockouts (RPL mutants, RPS mutants, *tsc1*, *tsc2)* were generated by mating the GapR-GFP query strain to strains with gene knockouts made with the *kanMX6* resistance gene for G418 resistance from the Bioneer deletion collection followed by random spore recovery. Strains with the correct genotype were isolated by auxotroph and drug selection, and gene deletion was confirmed by PCR. In some strains the kanamycin resistance cassette was replaced with the *hphMX6* hygromycin resistance cassette by transforming the G418 resistant strains with a PCR amplicon encoding hygromycin resistance (hygR).

The Rpa43-BFP plasmid was generated using pAV0471 from [58]. Rpa43 was tagged with mTagBFP2 and placed under constitutive expression by the tdh1 promoter. The plasmid was integrated into the ura4-DE locus of strains 646 and 1170 following digestion with AfeI, leaving the endogenous copy of rpa43 intact and unperturbed to support RNA pol I function.

## Induction of GapR-GFP for imaging and ChIP

GapR-GFP expression was induced by adding 100nM β-estradiol (stock in 100% ethanol). Cells were cultured in EMMS until log phase from a starting OD of 0.1–0.15. Cells were grown at 30˚C for 5–6 hours to allow for GapR-GFP induction.

## Confocal microscopy, FRAP, and FRAP analysis

Images were acquired through a Plan Apo 60x Objective (Nikon) on a Nikon Ti-E microscope fitted with a CSU-W1 (Yokogawa) spinning disc. GFP was excited at 488nm and mCherry was excited at 561nm. Emission for each was collected using an ET525/36m bandpass filter and an ET605/70m bandpass filter, respectively. Integration times for the Flash4 sCMOS (Hamamatsu) camera were typically 40-200ms and were adjusted as needed. When z stacks were acquired, a z spacing of 0.3μm was used. For timelapse imaging of WT cells expressing GapR-GFP and Nhp2-mCherry, images were acquired at 10-minute intervals with a z spacing of 0.5μm.

FRAP was conducted on a Perkin Elmer Ultraview Vox Spinning Disc using a 100x objective NA with a Hamamatsu C9100-23B EMCCD camera/detector. Single-place images were acquired. GFP was photobleached using a pulse of 488-nm laser power within a region of interest (ROI) dependent on strain genotype. For GFP-NLS cells, the ROI was the nucleus. For GapR-GFP cells, the ROIs were in the rDNA and bulk chromatin (non-rDNA). For FRAP analysis, the background was subtracted using the average intensity value of a ROI outside the cell. For every photobleached region, the recovery curve of average intensity was collected over 60 seconds (GFP-NLS cells) or 300 seconds (GapR-GFP cells). To correct for photobleaching during time-lapse acquisition, GFP-NLS ROI were normalized to intensity for the whole cell.

Similarly, GapR-GFP ROI (rDNA and bulk chromatin) were normalized to the intensity for the entire nucleus.

## Imaging of Rpa43-BFP

Images were acquired of WT and *rpl3801* mutant strains expressing Rpa43-mTagBFP2, GapR-GFP, and Nhp2-mCherry on a confocal spinning disc (Nikon Ti2 base with a Yokogawa CSU-W1). We found prolonged time in liquid medium perturbed Rpa43-BFP cellular localization. To circumvent this issue, we imaged cells as quickly as possible after suspension in liquid medium. To induce GapR-GFP expression for the glucose starvation experiment, single colonies were patched onto EMMS plates and then replica plated onto EMMS plates with 100nM β-estradiol overnight. The next day, cells were scraped from the EMMS+ β-estradiol plates into liquid medium with β-estradiol and placed in a Cellasic (Onix 2, Millipore) microfluidic chamber to allow fresh medium to continuously flow across the cells. For the glucose starvation experiment, strains were imaged after a 1.5 hour incubation in medium with or without glucose on the Cellasic chamber, with the temperature maintained at 25˚C. Both media had 100nM β-estradiol to maintain GapR-GFP expression. For the WT versus *rpl3801* mutant experiment, cells were directly inoculated from EMMS+ β-estradiol plates to liquid medium in the Cellasic chamber and immediately imaged.

Cells were imaged with a 60x Plan Apo (NA = 1.4) objective onto an sCMOS camera (Hamamatsu Flash 4). mTagBFP2 was excited at 405nm, while GFP was excited at 488nm and mCherry was excited at 561nm. Laser powers were 100% while the integration time for each channel was mCherry 50ms, GFP 50ms, and mTagBFP2 100ms. Transmitted light was also collected. During the acquisition, small z stacks of 3μm were collected with 0.5μm steps.

## Image processing

Representative images were processed using FIJI/ImageJ [59]. Quantitative image processing was automated using Python; Nikon image files were opened with the nd2reader (https://github.com/Open-Science-Tools/nd2reader) package into NumPy arrays [60]. Preprocessing fluorescent images before thresholding consisted of background subtraction and smoothing individual z-slices in 2D. For background subtraction, the OpenCV white top-hat function was used with a rectangular structuring element of side length 25 pixels [61]. The SciPy ndimage library was used to apply the Gaussian filter with a width of one sigma [62]. To limit the effects of nuclear orientation on our quantification, we generated three-dimensional nuclear and nucleolar masks using the Otsu threshold algorithm from scikit-image [63,64]. To account for variability in rDNA morphology during the cell cycle, we limited our analysis to cells in the G2 (late interphase) by filtering out cells with 2 nuclei. G2 cells were further separated into early or late G2 by cell length, where the shortest 33% of cells were called early G2 and the longest 33% of cells were called late G2. For quantification of the rDNA, we used the overlap of the 3D nuclear (GapR-GFP) and nucleolar (Nhp2-mCherry) masks. The mean intensity (arbitrary units) and volume (pixels$^3$) of each rDNA object (defined as GapR-GFP signal within the nucleolar mask) was measured and normalized by the mean intensity or volume of the nuclear object. These values were called the rDNA mean GFP intensity ratio (i.e. rDNA mean GFP intensity normalized to total nuclear mean GFP intensity) and the rDNA volume ratio (i.e. rDNA volume normalized to total nuclear volume). The nucleolar volume was also normalized to the nucleus and called the mCherry volume ratio (i.e. mCherry volume normalized to total nuclear GFP volume). Imaging quantification was conducted in Graphpad Prism 9.5.1 and R version 4.1.2. For plots produced in Graphpad Prism, which includes plots with single biological replicates, outliers were removed using the ROUT method (Q = 1). For

plots produced in R, outliers were defined as values 1.5 times outside the IQR and were removed. For every strain, 3 biological replicates were imaged in reference to WT or control strains.

For images of BFP tagged strains, fluorescence channels were segmented by a combination of peak finding and region growing. Each image was background subtracted with the OpenCV white tophat morphology function using a square structuring element 15 pixels wide. The image was smoothed using a gaussian filter with a sigma of 1 pixel. The threshold used for each channel was 2 times the mean of each channel. The GFP channel was eroded once and the BFP channel was eroded twice using the scikit-image binary erode function. Processing codes were uploaded to Github https://github.com/cwood1967/Cockrell_2024_PLoSGenetics.

## RNA purification

RNA was extracted using hot acid phenol-chloroform following [65]. Cultures were grown at 30˚C from a starting OD of 0.1–0.15 and collected after two doublings. Cells were pelleted at 2000-3000g for 2 min, liquid decanted, and then frozen on dry ice and stored at -70˚C prior to extraction. DNAse I treatment using NEB DNAse I RNAse-free enzyme was applied to 10mg of RNA, followed by acid-phenol chloroform extraction and ethanol precipitation to purify treated RNA.

## Dot blot analysis

30-50mL cultures of cells were grown until 2 OD doublings and labelled with 5mM 4-thiouracil (4-thiouracil resuspended in dimethylformamide) for 5 min. Cultures were either added to 20mL of ice-cold methanol or directly pelleted at 2000g for 2 min and snap frozen in liquid nitrogen. RNA was extracted (see RNA purification). RNA was biotinylated at a ratio of 10ug RNA to 1ug Biotin in 1x biotinylation buffer (10mM Tris-HCl pH 7.5, 1mM EDTA pH 8) at room temperature for 30 minutes in the dark. Biotinylated RNA was purified using acid-phenol purification.

To examine RNA by dot blot, 1uL volumes of RNA were spotted onto nylon positive membranes and baked for 30 minutes at 80˚C. Membranes were washed at room temperature for 2min in 2XSSC buffer with 0.1% SDS, followed by a 2min wash in 1XSSC buffer with 0.1% SDS. Membranes were incubated with LI-COR streptavidin IR-dye 800CW at a 1:10000 dilution in 1XSSC buffer with 0.1% SDS for 20min. Membranes were washed for 2x5 or 10 minute washes at room temperature in 2X SSC buffer with 0.1% SDS, followed by 2x5 or 10 minute washes at room temperature with 1X SSC buffer with 0.1% SDS. Membranes were dried for 5 minutes at 80˚C before imaging on the LI-COR Odyssey DLx imager.

Signal intensity was measured using Empiria Studio Software version 2.3. Background signal was measured in 3 separate areas on the membrane, the average of which was subtracted from the sample signal intensities. Plots depicting average signal intensity and standard deviation were generated in Graphpad Prism version 9.5.1.

## cDNA synthesis and qPCR

cDNA was synthesized using BioRad iScript gDNA Clear cDNA Synthesis kit. PCR primers are listed in S8 Table. For qPCR, each assay was setup in technical triplicate using automation by a Tecan EVO PCR workstation, and reactions were cycled and products detected using a QuantStudio 5 384-well Real-Time qPCR machine.

## Chromatin immunoprecipitation and qPCR

Chromatin immunoprecipitation was performed as described [66]. 50mL EMMS cultures were grown at 30˚C from a starting OD ~0.12 and allowed to grow for 1 hour, at which point 100nM β-estradiol was added to induce GapR-GFP expression. Cells were collected after two doublings. 5mL of cell culture was spun at 2000g for 2min, snap frozen, and stored at -80˚C for future RNA extraction and RNA sequencing. Cells were crosslinked with 1% formaldehyde for 15 min at room temperature then quenched with 125mM glycine for 5 min. Cells were washed 2x with ice-cold PBS, incubated in 0.4mg/mL zymolyase at 37˚C for 20 min, washed 2x in PEMS buffer, then pelleted and frozen on dry ice. Chromatin was sonicated using a Covaris s220 sonicator (PIP = 240W, DF = 20%, CPB = 200, time = 22min, running temperature = 6–10˚C, 1mL milliTUBE). 1mL of supernatant was transferred to a fresh tube following centrifugation. 500 microliters of supernatant was precleared using 25 microliters of Protein A dynabeads at 4˚C for 2 hours.

The supernatant was split into 350 microliters aliquots, each aliquot was incubated with 25 microliters protein A dynabeads and 1:1000 abcam Anti-GFP—ChIP Grade antibody (#AB290). Washes were performed as described [66]. Beads were pooled for elution, resuspended in elution buffer (50mM Tris-HCl, 1% SDS, 10mM EDTA), and incubated overnight at 65˚C. Supernatant was treated with RNAse A and proteinase K followed by purification using a Qiagen PCR purification kit.

For qPCR analysis, each sample was measured in biological triplicate using 2 microliters of IP and 2 microliters of 1:10 diluted input per sample. Primers for rDNA assays are provided (S8 Table). Each assay was performed in technical triplicate and reactions were cycled and products detected using a QuantStudio 5 384-well Real-Time qPCR machine. % input was calculated and normalized to the negative control locus for all samples.

## Generation of GapR-Bioneer screening library & validation

All pinning/arraying steps were carried out via the Singer RoTor Robot. The haploid *S. pombe* Bioneer deletion collection was arrayed onto YES containing plus-plates in 384-position format. Query strains were grown in flasks containing 100mL YES liquid media and then were arrayed onto YES plus-plates in 384 position format. For every 3 deletion mutant 384 plates, 1 384 query strain plate was made. Plates were para-filmed and grown at 30˚C for 2–3 days. Query strains were then pinned onto SPAS plus-plates (384). Deletion mutants were pinned directly on top of the query strains. To enhance mating efficiency, 384-long pin replicator pads were used to transfer a droplet of sterile water onto the mated cells and an "agar mix" function was utilized from the RoTor software to punch down several times onto the cells. SPAS 384 plate-plates were incubated at 26˚C for 3 days to induce sporulation. SPAS 384 plus-plates were incubated at 42˚C to eliminate haploids. SPAS 384 plus-plates were then pinned onto YES 384 plus-plates. SPAS 384 plus-plates were then pinned onto YNB-G418-cloNatR-hygR-ade-leu 384 plus-plates to select for double mutants. This step was then repeated. All 384 final selection plates were broken down into 96 well microtiter plates containing 200mL YELG418. Plates were cultured for two nights on a shaking incubator at 30˚C and were then made into glycerol stocks.

For the validation screen, the above approach was used on a subset of hits identified from the initial screen (S3 Table). This sub-collection was mated to a prototrophic query strain (GS1138, S6 Table) following the protocol described above.

## Imaging screen

Strains were inoculated from glycerol stocks into liquid YEASupG418 culture and grown at 30˚C for 2 days. Cells were diluted to an OD of 0.1 in EMMS and grown overnight at 25˚C.

The next morning, cells were diluted 1:20 in fresh EMMS medium with 100nM β-estradiol and grown for 6 hours at 30˚C. Cells were aliquoted into 384-well poly-L-lysine coated imaging plates. Plates were briefly centrifuged at 1000g for 5 min at room temperature and then imaged on an Opera Phenix microscopy system (Perkin Elmer). A 63x water immersion objective (NA 1.15) was used to acquire z stacks of 3 images with 1μm spacing. EGFP, mCherry and transmitted light images were acquired at each z slice with appropriate lasers and filters. Specifically, GFP was excited at 488 nm and mCherry was excited at 561nm. GFP was collected through a 500-550nm bandpass filter while mCherry was collected through a 570-630nm bandpass filter. 4 fields of view were imaged per well which resulted in hundreds of yeast imaged per strain.

## Screen data processing and analysis

Image quantification was conducted as described in "Image processing." The resulting data underwent several filtering steps to determine rDNA phenotypes in comparison to WT controls. First, empty wells were identified as having low GFP intensity values and extremely high or low GFP volume. Datapoints were filtered out that had GFP intensity values above 750 and GFP volume between 10 and 200. Median values were then determined for the two mated duplicates (each 384-well plate contained two 96-well plates from the Bioneer collection that were mated to the QS to account for variability from mating). Wells with fewer than 30 datapoints (i.e. cells) were filtered out. The remaining datapoints represented the median value of two biological replicates per mutant strain.

Each 384-well plate contained control strains in quadruplicate (4 wells per strain). These included WT (query strain) and several positive controls for significantly different values from WT (S10 Fig). To generate a threshold for WT rDNA values, median rDNA GFP intensity and rDNA volume values from WT strains on plates 1, 2, 4, 7, 11, 12, 15, and 16 were averaged. These plates were selected because all 4 control strains behaved as expected. The controls on the remaining plates showed variability in 1 or more controls and were not used for thresholding. The threshold for hits with increased rDNA GFP and increased rDNA volume was set as the WT averaged threshold plus one-third the interquartile range. The threshold for hits with decreased rDNA GFP and decreased rDNA volume was set as the WT averaged threshold minus one-fourth the interquartile range. These thresholds set a reasonable number of hits (123 high hits and 192 low hits) for additional validation and analysis.

To look for pathway enrichment, the 123 high hits and 192 low hits were individually run through the STRING Network Analysis database version 11.5. Settings included: full STRING network, all active interaction sources were selected, and a minimum required interaction score of 0.7 (high confidence) was used. MCL clustering with an inflation parameter of 2 was used with dotted lines indicating the edges between clusters. The 192 low hits failed to show functional enrichment and were not investigated further.

## Synthetic genetic analysis

Query strains with hygR were mated to candidate G418 resistant strains as described in "Generation of GapR-Bioneer library". Query strains included 5 RPL gene deletion strains with Torin1 resistance. 1 RPL and 2 RPS gene deletion strains that lacked Torin1 resistance served as controls. Strains were inoculated from glycerol stocks into liquid YEASup culture and grown at 30˚C for 2 days. Cultures were pinned onto EMMS plus plates without drug, with DMSO control, or with 12.5 micromolar Torin1 inhibitor using a Singer RoTor Robot. Plates were grown at 30˚C and imaged daily for 5 days. To quantify growth of each mutant, we measured colony size, or density, in ImageJ/FIJI (S5 Table). Empty wells were often assigned low

or negative colony densities; to account for this, all negative colony densities as well as the bottom 5% of positive colony densities were converted to "1" to represent no growth. We calculated the fold change in growth as the colony density of each double mutant on 12.5μM Torin1 divided by colony density on DMSO control plate. To better visualize differences in sensitive and resistant strains, fold changes were scaled by $\log_2$. A double mutant was sensitive to Torin1 if no growth, or a fold change less than negative 10, was calculated. A double mutant was resistant to Torin1 if growth, or a fold change greater than negative ten, was calculated. If a candidate gene deletion led to sensitivity to Torin1 in 3 out of 5 RPL query strains, that candidate gene was considered essential for Torin1 resistance. If a candidate gene deletion led to Torin1 resistance in 2 out of 3 control strains, that candidate gene was considered a repressor of Torin1 resistance.

## Digital droplet PCR

Genomic DNA was extracted and quantified as previously described [26]. Each 20 microliter reaction contained a total of 0.005 ng of gDNA, restriction enzyme MseI, BioRAD master mix for probes (no dUTP), primers and probes for 28S and *nda2* sequences listed in S8 Table. Each assay was performed in technical triplicate. Strains were considered to have approximately equal 28S copy number if they were within 10% (plus or minus) of the WT 28S copy number.

## RNA-Seq

Cells were collected during preparation for chromatin immunoprecipitation experiment (see Chromatin immunoprecipitation and qPCR). 3 biological replicates of WT (strain 646, S6 Table) cultures were grown in EMMS at 30˚C for 2 doublings. Cell pellets were collected and followed by RNA extraction as described in "RNA purification."

mRNAseq libraries were generated from 500 ng of high-quality total RNA, as assessed using the Bioanalyzer (Agilent). Libraries were made according to the manufacturer's directions for the TruSeq Stranded mRNA Library Prep kit- 48 Samples (Illumina), and TruSeq RNA Single Indexes Sets A and B (Illumina). Resulting short fragment libraries were checked for quality and quantity using the Bioanalyzer (Agilent) and Qubit Fluorometer (Life Technologies). Libraries were pooled, requantified and sequenced as 75bp paired reads on a mid-output flow cell using the Illumina NextSeq 500 instrument. Following sequencing, Illumina Primary Analysis version RTA 2.4.11 and bcl2fastq2 v2.20 were run to demultiplex reads for all libraries and generate FASTQ files.

## Supporting information

**S1 Fig. GapR-GFP binds to rDNA in a transcription-dependent manner.** ChIP-qPCR of GapR-GFP enrichment at the rDNA locus is decreased by glucose starvation. Amplicon positions are marked by black bars. The bars show the mean %IP for the replicates with error bars depicting standard deviation. Statistical significance determined by unpaired t-test comparing glucose treated versus starved cells is shown with ns = not significant, *p<0.05, and **p<0.01 (EPS)

**S2 Fig. Characterizing the effect of GapR-GFP expression on rDNA function and cell health.** (A) Total nascent RNA levels were assessed in GapR-GFP and GFP-NLS expressing strains by RNA spot assay. Nascent RNA was pulse labeled for 5 minutes by adding 4-thiouracil to log-phase cultures. RNA was extracted and biotinylated. Each spot represents 1 μL of biotinylated RNA at 200ng/μL, 150ng/μL, 100ng/μL, and 50ng/μL concentrations. Nascent RNAs were detected by incubation with a streptavidin-IR dye. (B) Quantification of streptavidin-IR

signal intensity for labeled nascent RNAs from panel A. Each bar represents the average of 3 biological replicates, with overlying dots representing the 3 replicate values. Error bars show standard deviation. Statistical significance was determined by unpaired t-test where ns = not significant. (C) Plot shows qPCR analysis of rRNA steady state transcripts in GFP-NLS and GapR-GFP cells with β-estradiol induction. rRNA transcripts were normalized to housekeeping gene pyk1. GFP-NLS samples were normalized to 1 for comparison with GapR-GFP transcript levels. (D) Cell cycle staging of GapR-GFP and GFP-NLS expressing cells is shown. For each strain, 100 cells per biological replicate were classified as G2, S phase, or mitotic. Error bars represent the standard deviation for 3 biological replicates. (E) A growth assay comparing GapR-GFP and GFP-NLS strains with or without β-estradiol induction is shown, with cells grown at 30˚C for 24 hours. Growth was measured every 15 minutes from a starting $OD_{600}$ = 0.01. Each line depicts the average growth of 3 biological replicates. (F) Nucleolar (mCherry) volume is normalized to cell length in late G2 cells for GFP-NLS and GapR-GFP strains with median values for 3 biological replicates shown. Color represents biological replicate. Significance for the average of the 3 median values was determined by unpaired t-test with ns = not significant.
(EPS)

**S3 Fig. FRAP analysis of GapR-GFP shows no protein turnover.** (A) Plot shows the average fluorescence recovery after photobleaching for WT cells in G2 expressing GapR-GFP (n = 7), with similar recovery observed for GapR-GFP at the rDNA loci and bulk chromatin. Error bars represent standard deviation. (B) Plot shows the average fluorescence recovery after photobleaching for WT cells in G2 expressing GFP-NLS (n = 3).
(EPS)

**S4 Fig. rDNA spatial organization reflects 28S copy number in cells in early or late G2.** All plots show quantification of rDNA volume ratio (nucleolar GapR-GFP to total GapR-GFP), the GFP intensity ratio (nucleolar GapR-GFP to total GapR-GFP), or nucleolar volume (nucleolar Nhp2-mCherry to total GapR-GFP) for cells in early G2 (A-C) or late G2 (D-F) with the copy number indicated (58, 80, or 147). Each plot shows 3 biological replicates per strain, with a distinct color per replicate. Median values for each biological replicate are plotted as larger circles. Statistical significance was determined by unpaired t-test, where ns = not significant, $*p<0.05$, $**p<0.01$, and $***p<0.001$.
(EPS)

**S5 Fig. 4-thiouracil incorporation is not detected in glucose starvation.** (A) Total nascent RNA levels were assessed in WT cells with and without glucose starvation for 2 hours by RNA spot assay. Each spot represents 1μL of biotinylated RNA at 200ng/μL derived from RNA extracted from cells following a 5 minute 4-thiouracil pulse labeling. (B) Quantification of streptavidin-IR labeled nascent RNA levels from panel A is plotted for glucose treated and glucose starved WT cells. Each bar represents the average of 3 biological replicates, with overlying dots representing the 3 replicate values. Error bars depict standard deviation. Statistical significance was determined by unpaired t-test where $**p<0.01$.
(EPS)

**S6 Fig. rDNA spatial organization in cells in early or late G2 with glucose or starved for glucose.** All plots show quantification of rDNA volume ratio (nucleolar GapR-GFP to total GapR-GFP), the GFP intensity ratio (nucleolar GapR-GFP to total GapR-GFP), or nucleolar volume (nucleolar Nhp2-mCherry to total GapR-GFP) for cells in early G2 (A-C) or late G2 (D-F). Each plot shows 3 biological replicates per strain, with a distinct color per replicate. Median values for each biological replicate are plotted as larger circles. Statistical significance

was determined by unpaired t-test, where ns = not significant, $^*p<0.05$, $^{**}p<0.01$, and $^{***}p<0.001$.
(EPS)

**S7 Fig. Inhibition of RNA pol I results in relocation of rDNA arrays to the nucleolar periphery.** (A) Images for cells in G2 treated with 25 μg/μl Actinomcyin D or DMSO for 120 minutes are shown. The scale bar represents 1μm. The intensity view of the mCherry signal (cobalt) highlights Nhp2 dispersal throughout the nucleus. (B-D) Plots showing quantification of rDNA and nucleolar morphology for DMSO control and actinomycin D treated WT cells in late G2 imaged after 120 minutes of treatment. The rDNA volume ratio (B), rDNA mean GFP intensity ratio (C), and mCherry volume ratio (D) for 3 biological replicates are plotted with a distinct color per replicate. The median values of 3 biological replicates are shown. Statistical significance of the 3 biological replicates was determined by unpaired t-test, where ns = not significant, $^*p<0.05$, and $^{**}p<0.01$.
(EPS)

**S8 Fig. Alternate rDNA marker Rpa43-BFP forms nucleolar foci with glucose starvation.** (A) Representative images for early and late G2 cells in EMMS media with or without glucose for 1.5 hours are shown. The scale bar represents 1μm. (B) Plots showing quantification of Rpa43-BFP morphology in WT cells in G2 phase after 1.5 hours of glucose starvation. All G2 stages were included due to low cell number in glucose starved conditions. The Rpa43-BFP volume is normalized to total nuclear volume using GapR-GFP (n>11 cells). Two biological replicates are shown. Bars represent the median and interquartile range. Statistical significance of each replicate was determined by unpaired t-test, where $^{***}p<0.001$ and $^{****}p<0.0001$.
(EPS)

**S9 Fig. rDNA spatial organization in cells in early or late G2 in TOR mutants.** All plots show quantification for cells in early G2 (A-C) or late G2 (D-F). Each plot shows 3 biological replicates per strain, with a distinct color per replicate. Median values for each biological replicate are plotted as larger circles. Statistical significance was determined by unpaired t-test by comparing each mutant to WT, where ns = not significant, $^*p<0.05$, and $^{**}p<0.01$.
(EPS)

**S10 Fig. Strains with deletions impacting GapR-GFP for use as positive controls in the imaging screen.** Strains were identified with rDNA phenotypes significantly different from WT. These strains were included on every imaging plate. Representative data from imaging screen plate 1 is shown, plotting (A) rDNA volume and (B) rDNA mean GFP intensity. Bars represent the median and interquartile range. Statistical significance was determined by the Kruskal-Wallis test (one-way ANOVA) with comparison to WT, where $^{****}p<0.0001$.
(EPS)

**S11 Fig. Gene groupings from STRING analysis for 123 "high" hits.** Nodes in yellow indicate genes that were positive for increases in both rDNA volume and mean GFP intensity following validation. Nodes in gray indicate genes that were positive for increases in mean GFP intensity following validation. Nodes in white did not validate for either rDNA parameter.
(EPS)

**S12 Fig. rDNA spatial organization in RP mutant cells in early and late G2.** All plots show quantification for cells in early G2 (A-C) or late G2 (D-F) G2. Each plot shows 3 biological replicates per strain, with a distinct color per replicate. Median values for each biological replicate are plotted as larger circles. Statistical significance was determined for each RP mutant relative to WT by unpaired t-test. Statistical significance is noted by an asterisk, where $^*p<0.05$ and

**p<0.01.
(PDF)

**S13 Fig. Alternate rDNA marker Rpa43-BFP detects rDNA spatial organization phenotype in RPL mutant.** Plots quantifying rDNA volume using (A-B) GapR-GFP or (C-D) Rpa43-BFP for WT and *rpl3801Δ* mutant strains in early and late G2 are shown. For both the GFP and BFP markers, rDNA volume was normalized to total nuclear volume using GapR-GFP. Each plot shows 3 biological replicates per strain, with a distinct color per replicate. Median values for each biological replicate are plotted as larger circles. Statistical significance was determined by unpaired t-test, where**p<0.01.
(EPS)

**S14 Fig. An RPL mutant with an rDNA organization phenotype does not have an altered level of nascent RNA.** (A) Total nascent RNA levels were assessed in WT or *rpl3801Δ* mutant cells following a 5 minute 4-thiouracil pulse labeling by RNA spot assay. Each spot represents 1μL of biotinylated RNA at 200ng/μL, 150ng/μL, 100ng/μL, and 50ng/μL concentrations. Nascent RNAs were detected by incubation with a streptavidin-IR dye. (B) Quantification of streptavidin-IR signal intensity for labeled nascent RNAs from panel A is shown. Each bar represents the average of 3 biological replicates, with overlying dots representing the 3 replicate values. Error bars depict standard deviation. Statistical significance was determined by unpaired t-test where ns = not significant.
(EPS)

**S15 Fig. GapR-GFP binding profiles and rDNA organization in an RPL and an assembly mutant.** (A) ChIP-qPCR of GapR-GFP binding at rDNA locus in RPL mutants is shown. qPCR amplicons are represented by black bars above the rDNA locus. The bars show the mean %IP for the replicates with error bars depicting standard deviation. Values above the bars indicate the p-value determined by unpaired t-test comparing RP mutant to WT, with ns = not significant. (B-D) Quantification of rDNA spatial organization and nucleolar phenotypes in cells in late G2 for *rpl3602Δ* and *mrt4Δ* strains is shown. Plots show 3 biological replicates for (B) rDNA volume, (C) rDNA mean GFP intensity, and (D) mCherry volume, with distinct colors per replicate. The median values of 3 biological replicates are shown as overlying dots. Statistical significance of the 3 biological replicates was determined by unpaired t-test comparing RP mutants to WT with ns = not significant, **p<0.01, ***p<0.001, and ****p<0.0001.
(EPS)

**S16 Fig. SGA for modifiers of Torin1 resistance in RP mutants.** A heatmap depicting the fold change in growth on Torin1 inhibitor for RP and ribosome biogenesis signaling pathway double mutants is shown. The Y-axis labels 110 genes from candidate ribosome biogenesis signaling pathways. The X-axis labels the query strains including 5 Torin1 resistant RPL strains (left) and 3 Torin1 sensitive RPL and RPS strains (right). Values plotted include the fold change in growth between medium containing 12.5μm Torin1 or containing DMSO as a control, shown in $\log_2$. Double mutants with a $\log_2$ fold change ($\log_2 fc$) below -10 are sensitive to Torin1, while double mutants with a $\log_2$ fold change above -10 are resistant to Torin1. Gray boxes represent double mutants where growth could not be measured on both Torin1 and DMSO media.
(EPS)

**S1 Table. of rDNA quantification ratios and hits from the initial rDNA imaging screen.**
(CSV)

**S2 Table. of strains absent from the initial rDNA imaging screen.**
(CSV)

**S3 Table. of rDNA quantification ratios and hits from the validation rDNA imaging screen.**
(CSV)

**S4 Table. of mRNA expression between ribosome protein paralogs.** The gene name for the first paralog is listed. Columns 2–4 list the average TPM of 3 biological replicates for the first, second, and third paralogous genes. The color scales from low (blue) to high (yellow) expression. *RP strains with the high rDNA activity phenotype including 6 strains from the imaging screen as well as the *rpl3602* control strain.
(XLSX)

**S5 Table. of colony density values and fold change for the SGA screen on torin1.**
(CSV)

**S6 Table. of yeast strains used in study.**
(CSV)

**S7 Table. of plasmids used in study.**
(CSV)

**S8 Table. of primers and probes used in study.**
(CSV)

## Acknowledgments

We thank members of the Gerton lab for constructive feedback and suggestions on the experiments included in this manuscript, particularly Dr. Tamara Potapova for assistance with the FRAP assay, Dr. Joe Varberg for assistance with data presentation, and Dr. Devika Salim for ddPCR primer design. We thank Dr. Ilya Finkelstein, Dr. Susan Forsburg, Dr. Sarah Zanders, and Dr. Jose Cansado for sharing *S. pombe* strains. This work contributes to the requirements for AJC's PhD thesis research at the University of Kansas School of Medicine.

## Author Contributions

**Conceptualization:** Alexandria J. Cockrell, Jennifer L. Gerton.

**Data curation:** Alexandria J. Cockrell, Jeffrey J. Lange, Christopher Wood.

**Formal analysis:** Alexandria J. Cockrell, Jeffrey J. Lange, Christopher Wood.

**Funding acquisition:** Alexandria J. Cockrell, Monica S. Guo, Jennifer L. Gerton.

**Investigation:** Alexandria J. Cockrell, Jeffrey J. Lange, Christopher Wood, Mark Mattingly, Scott M. McCroskey, William D. Bradford, Juliana Conkright-Fincham, Lauren Weems.

**Methodology:** Alexandria J. Cockrell, Jeffrey J. Lange, Christopher Wood, Mark Mattingly, Monica S. Guo, Jennifer L. Gerton.

**Project administration:** Alexandria J. Cockrell, Jennifer L. Gerton.

**Resources:** Alexandria J. Cockrell, Monica S. Guo, Jennifer L. Gerton.

**Software:** Jeffrey J. Lange, Christopher Wood.

**Supervision:** Alexandria J. Cockrell, Jennifer L. Gerton.

**Validation:** Alexandria J. Cockrell, Jeffrey J. Lange, Christopher Wood, Mark Mattingly, Scott M. McCroskey, William D. Bradford.

**Visualization:** Alexandria J. Cockrell, Jennifer L. Gerton.

**Writing – original draft:** Alexandria J. Cockrell, Jennifer L. Gerton.

**Writing – review & editing:** Alexandria J. Cockrell, Jeffrey J. Lange, Christopher Wood, Scott M. McCroskey, William D. Bradford, Juliana Conkright-Fincham, Lauren Weems, Monica S. Guo, Jennifer L. Gerton.

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
