## [Decision Letter · Decision Letter 0]

10 Jul 2023

Brussels, July 5, 2023

Dear Dr Gerton,

Dear Jen,

Your work entitled 'Regulators of rDNA morphology in fission yeast'  submitted to *PLoS Genetics *has now been carefully evaluated by four experts in the field of ribosome biogenesis. I have also read the paper with interest.

While all four referees, and myself, recognize the timeliness and originality (use of an heterologous reporter, live cells, etc.) of your approach and the importance of the topic (how nucleolar morphology reflects function), they all raised substantial concerns that would need to be fully addressed in a revised version of the manuscript.

Personally, I also found the approach very original, but I agree that to prove that GapR-GFP does not interfere with Pol I function, there is a need to perform a pulse-chase/metabolic labelling (or EM spreads). Also, I often thought, is Nhp2 really the best marker here? Could some results be confirmed with another nucleolar protein co-staining? Would it be possible to confirm the change in rDNA organization by use of an endogenous protein detection (eg an rDNA binder) or by some other means?

It will be important also to explain well how rDNA and nucleolar volume were calculated, and how the effects of cell-cycle were taken into account (were cells synchronized?). A few excellent recommendations on semantics were suggested, which will also improve the paper.

Based on their overall positive assessment and constructive criticisms, I would like to suggest you prepare a revision of your manuscript. Since this will require the addition of novel experimental work, I would like to suggest an initial 6-month deadline for resubmission. Feel free to contact me if you need to discuss your revision plan, or if you need an extension.

Best wishes,

Denis

We are sorry that we cannot be more positive about your manuscript at this stage. Please do not hesitate to contact us if you have any concerns or questions.

Yours sincerely,

Denis LJ LaFontaine, PhD

Guest Editor

PLOS Genetics

Eva Stukenbrock

Section Editor

PLOS Genetics

Reviewer's Responses to Questions

**Comments to the Authors:**

Reviewer #1: referee#1's report was posted as an attachment and should be downloaded

Reviewer #2: The manuscript entitled " Regulators of rDNA morphology in fission yeast" by Cockrell et al., explore the ribosomal DNA structure using a bacterial protein, GapR, known to associate with overtwisted DNA. Authors characterize GapR binding in vivo by quantitative imaging, and could show that GapR can be used as a very useful reporter construct to investigate rDNA structure in vivo. rDNA modification during cell cycle, upon transcriptional repression, or TOR deregulation is of interest. Using this new reporter, they were able to screen an entire library of haploid deletion, which is clearly a master piece of this work. Due to frequent variation of rDNA copy number which could affect screen results, authors focused on 118 candidates to validate screen results. They could identify a very interesting effect of 60S biogenesis defect on rDNA morphology

The following part of the manuscript is less focused, with some connections with signaling pathway which are not always conclusive. Some results presented here appears essentially preliminar, such as Synthetic Genetic Array to explore signalling pathways altered in 60S mutants (presented in figure 5B). Global interaction presented here are not directly demonstrating authors statement (Our genetic screen indicated all assayed signaling pathways can provide input into Torin1 resistance). Figure 6 depict a possible connection with Cell Integrity MAPK pathway which appears more robust.

Overall, this study is very interesting and provide some important insight on rDNA structure in vivo. Some important control are required, and extensive editing is required. Numerous data are presented the text, in figures and supplementary figure that are not always crucial (see SGA screen). Authors should clarify the text, and select required figures.

Major critique:

GapR expression has no obvious negatively impact on rDNA function or cell health. Authors should also check that rDNA copy number is not impacted using either qPCR, or Pulse-field gel electrophoresis (or digital droplet PCR as in figure 1C).

What about the ratio transcribed / silenced genes in rDNA of . pombe ? When the number of genes increase, does the ratio of active genes/silenced rDNA change? This analysis using psoralen cross-linking would be useful to explore if GapR binding is specific of transcribed genes.

Authors showed that GapR expression does not affect accumulation of rRNA or pre-rRNA (S2A). However, accumulation of rRNA can be equally affected by transcription, transcript stability and maturation rate. Authors should perform pulse labelling experiment to establish that in vivo transcription is not affected by GapR expression. Pulse labelling with 5mM 4-thiouracil is described in the method section, but is revealed using total RNA detection. In such detection, it is unclear which RNA species is detected. Why not using such thiouracil labelled RNA and analyze pre-rRNA, or rRNA labelled during such 2 min using immobilized probe, similar to Run-on assay (for budding yeast, see wery et al., RNA, 2009).

Minor critique:

In introduction, known re-organization of rDNA along cell cycle should be mention, to introduce the notion that rDNA reorganization depend for sure of transcriptional activity but also on cell cycle.

For imaging, wide field micrography to see the distribution of patterns of GapR-GFP among a yeast population would be very usefull.

GapR(1-76) fuse to GFP should be shown as comparison to the very clear rDNA binding of full-length GapR.

In budding yeast rRNA production is decreased during mitosis. Here, authors observed GapR binding in mitosis. What about the transcription level in S.pombe mitosis ?

Figure S5 is unclear and should be corrected; panel A is not informative, panel B is only showing total RNA, and not a transcriptional shut-off with 2 hours of glucose starvation (S5 Fig). (Please correct spelling in figure legend biontinylated RNA)

Figure S6. Authors should rephrase interpretation “Actinomycin D treatment causes Nhp2 diffusion throughout the nucleus (S6A Fig)”. An increase nucleoplasmic signal is observed, but a very large fraction of Nhp2 remains nucleolar. This should be mentioned.

In the screen, some expected hits are present (SPBC11G11.05, or sp-Rpa34) but other such as sp-Hmo1 (hmo1 / SPBC28F2.11) are not present in the table. This is surprising. Why in figure 3C, gene SPAC607.02c is named SPAC607.02c.1 ? Authors should mention which mutant is present, or absent from the deletion collection analyzed here.

Some results are confusing (While we observed a trend towards increased Pmk1 protein levels, this trend was not statistically significant (S11 Fig)). If not conclusive, this should not be shown. Alternatively, experiments should be repeated to strengthen the tendency authors discuss here.

Some small editing are required (“5/12/2023 4:22:00 PM” present page 6 lane 114)

Careful references editing should be made for example:

in ref 3, journal is missing : (Derenzini M, Trere D, Pession A, Montanaro L. Nucleolar Function and Size in Cancer Cells. 804 1998;152:7 )

in ref 19, a copy/paste when wrong, Berger, Barkai and editors are not authors of this work. (Guo MS, Kawamura R, Littlehale ML, Marko JF, Laub MT. High-resolution, genome-wide mapping of positive supercoiling in chromosomes. Berger JM, Barkai N, editors. eLife. 2021;10:e67236. )

Reviewer #3: This ms submitted to PLoS Genetics by Cockrell et al entitled “Regulators of rDNA morphology in fission yeast” takes a unique approach to discovering factors required for maintaining the rDNA arrays in a yeast that divides, S. pombe. They have pioneered the use of the GapR protein from Caulobacter that binds to twisted DNA, including in the rDNA repeats, to delineate the spatial organization of the rDNA in live cells. GapR was modified with S. pombe codon optimization, fused to GFP for detection, and placed under the control of an estradiol promoter in fission yeast. The basic assays that they use to report rDNA spatial organization are rDNA volume (from Nhp2-Cherry) and rDNA mean GFP intensity (from GapR-GFP). They screened for nonessential genes that when deleted affected these parameters and found many RPL proteins. Surprisingly, the observed phenotype, based on the GapR-GFP signal, was consistent with increased ribosome biogenesis. To define the signalling pathways that contribute to this, they used Synthetic Genetic Arrays and found that RPL insufficiency results in activation of the Pmk1 pathway. This study is an exciting approach, carefully done, that combines microscopy with genetics in S. pombe to probe ribosome biogenesis, revealing new signalling pathway inputs.

At the same time, I think there are some things that could be improved prior to publication.

1. The paper unfortunately relies on inference for some of the major conclusions and could be strengthened by some simple biochemical experiments. For example, increased ribosome biogenesis in the delta RPL strains is not tested directly but could be. Furthermore, decreased RPL levels is not tested at the level of protein, but could be. It would be important to know if these biochemical results concur with the microscopy, especially since this is the first use of this reporter gene.

2. There is a prevalent use of jargon that renders the ms difficult to understand. These are some examples where jargon is used and muddles the meaning. “60S ribosome subunit gene deletion” “our screen identified 60S mutants” “60S hits” It is all through the ms. I do not think that the RPL mutations should be described in this way. “60S ribosome subunit gene deletion” would mean to me deletion of the large subunit rRNAs, not the large ribosomal subunit proteins. A better way to say “our screen identified 60S mutants” is to say “our screen identified mutations in proteins of the large ribosomal subunit.” “60S hits” should be “RPL hits”. It is critical to avoid the use of shorthand, confusing jargon and refer to the RPL protein themselves, after first defining them as protein components of the large ribosomal subunit.

3. I must not understand the results in Fig 5, as I see variable torin resistance among the strains. Please state what was used as criteria with each concentration. For 10 microM, perhaps any growth in the 3rd dilution counted as resistance? If you instead used growth in all 5 dilutions, only one would be resistant. Perhaps I missed this logic, please explain.

4. While the ms relies on live cell signals from two reporter genes, a calculation is made in some of the figures that is related to them but is not defined that I can see. What is the rDNA volume ratio vs the rDNA volume? I don’t understand the ratio part—ratio of what? Please define and put into the text and methods.

5. The Discussion could be improved with greater attention to the published details. I do not agree with the interpretation that the haploinsufficiency of the bone marrow failure syndrome ribosomopathies acts as the oncogenic trigger. Indeed, because these RPS and RPL mutations and deletions cause bone marrow failure, they have reduced ribosome biogenesis, not activated as is found here. Furthermore, ongoing longitudinal studies on patient cells is revealing mutant p53 and oncogene activation as the causes for cancer predisposition.

6. Also, perhaps the Discussion could be strengthened by comparing the results here to the work of Nancy Hopkins in zebrafish PMC406397.

Typos: Line 114. Irrelevant writing.

Line 583: “Septation index” should be bolded.

Reviewer #4: Synthesis of ribosomes is a major energy consuming process in the cell and is tightly regulated according to different physiological conditions. Thus, it is discussed that cells employ different pathways to ensure a balanced up- and downregulation of the structural ribosomal components. One of the first steps of ribosome biogenesis is ribosomal (r)DNA transcription producing rRNA from multicopy tandemly repeated genes in a special nuclear compartment, the nucleolus. In the nucleolus the rRNA undergoes co-transcriptional folding and processing, and it is known that nucleolar structures change in good correlation with the activity of the rRNA synthesizing and maturing machineries. Clearly, a deeper understanding of the interrelationship between rDNA morphology and the regulation of ribosome biogenesis and signaling pathways is lacking.

In this work, the authors express the bacterial DNA-binding protein GapR in fusion with GFP in fission yeast as a marker for both spatial rDNA organization and increased rDNA condensation. GapR was previously shown to bind preferentially to transcriptionally induced overtwisted DNA. Accordingly, GapR-GFP binding should reflect changes in rDNA topology which might be provok

---

## [Decision Letter · Decision Letter 1]

25 Apr 2024

Dear Dr Gerton,

Thank you very much for submitting your Research Article entitled 'Regulators of rDNA morphology in fission yeast' to PLOS Genetics.

The manuscript was fully evaluated at the editorial level and by independent peer reviewers. The reviewers appreciated the attention to an important problem, but raised some substantial concerns about the current manuscript. Based on the reviews, we will not be able to accept this version of the manuscript, but we would be willing to review a much-revised version. We cannot, of course, promise publication at that time.

If you decide to revise the manuscript for further consideration at PLOS Genetics, please aim to resubmit within the next 60 days, unless it will take extra time to address the concerns of the reviewers, in which case we would appreciate an expected resubmission date by email to plosgenetics@plos.org.

We are sorry that we cannot be more positive about your manuscript at this stage. Please do not hesitate to contact us if you have any concerns or questions.

Yours sincerely,

Denis LJ LaFontaine, PhD

Guest Editor

PLOS Genetics

Eva Stukenbrock

Section Editor

PLOS Genetics

Dear Dr Gerton,

Dear Jen,

Please let us first apologize for the delay in getting back to you.

We have now received reports on your revised manuscript from three expert referees.

All three referees concur that a substantial amount of work has gone into the revision.

As you will read yourself, their recommendations are very contrasted.

Referee #2 is well satisfied and suggests only minor (mostly formatting) changes.

Referees #1 and #4 are still not truly convinced by several aspects.

An important and somehow shared criticism that remains is the lack of definitive demonstration that the GapR-GFP probe binds exclusively to transcriptionally induced overtwisted rDNA.

We really appreciate the amount of work you, and your team, have put into this submission, and we still believe your work has potential for PLOS GENETICS.

Therefore, we suggest you carefully review the points raised by reviewers #1 and #4.

If you believe you can address them satisfactorily and provide decisive information on the remaining critiques (or refute them efficiently, for example, because you don’t think they impact your work), then we would be happy to receive rapidly a revision. Otherwise, we suggest it is may better to transfer to another journal for the sake of efficacy and a timely publication of your work.

Kind regards

Denis

Reviewer's Responses to Questions

**Comments to the Authors:**

Reviewer #1: The authors have put considerable effort into revising their manuscript. It is intriguing and original. There still are problems with the primary data.

Regarding the data collection:

• The key fluorescent probe (GapR-GFP) is thought to bind “overtwisted DNA,” but is that especially relevant to rDNA ? and – if so – why ? It sounds as though the authors turned to GapR-GFP since staining with Hoechst was not reproducible, not because GapR-positivity implies that the DNA has some interpretable characteristic.

• All experiments depend on the addition of the GapR-GFP inducer, estradiol, for 5-6 hr (as opposed to steady-state). Is it reasonable to think that the many conditions that they study do not affect the rate or stability of induction ? If induction is variable, certainly any measurements of signal intensity are questionable.

• The data depend critically on the authors ability to quantitate the “area occupied by rDNA.” On the basis of what is shown – although much of the analysis seems to have been automated and shows some “statistical significance” – it is not possible to know how accurate those measurements are. Some examples also seem to contradict their graphs. Thus, for “representative” wt cells in Figure 4E, the Nhp2 signal is obviously small by comparison to Figure 1A/B. Why is this ??

• The authors now assure us that cell cycle progression does not interfere with their measurements and that the data pertain to G2. Since they refer to “early” and “late” G2, these time intervals should be designated in Figure 2B/C.

• The authors compare the performance of GapR to a mutant that is said not to bind to overtwisted DNA in Figure 1B. The mutant protein does not highlight the rDNA arrays but, as shown, it does primarily localize to the nucleolus. The authors offer no further comment, but the only DNA sequences that are known to be in the nucleolus are rDNA sequences. Might this “other” accumulation of GapR influence the estimates of “rDNA volume” ? This depends critically on the definition of the masks that were used for image analysis, but the reader has no way to know whether the masks were applied appropriately.

• The authors should compare their “hits” to those identified by Perrimon (their reference No. 11).

Reviewer #2: The revised manuscript entitled " Regulators of rDNA morphology in fission yeast" by Cockrell et al., explore the ribosomal DNA structure using a bacterial protein, GapR, known to associate with overtwisted DNA.

Overall, this revised study is very interesting and provide some important insight on rDNA structure in vivo. This revised version is extensively modified, and improved. Some results and interpretation have been clarified, or removed. Most required experiments have been performed, or carefully addressed. Some minor corrections would be of interest.

Major critique:

none

Minor critiques:

Size and orientation of figures is not homogeneous. This is not the best to evaluate data.

Supplementary figure (.eps format) are not formatted, with variable size, resolution, length. This should be corrected.

Legend of supplementary figures need some final corrections:

In all figures, significant difference (0.01 or 0.05) should be highlighted.

S4 fig : please indicate in figure legend quantification of what.

S5 fig : “Glucose starvation causes complete transcription shut-off”. This conclusion appears over-stated without a simple control. The assay rely on uptake of 4-thiouracil pulse labelling. Please indicate how uptake of 4-thiouracil was evaluate in glucose starvation? Other studies supporting this conclusion using other assay would also be of interest.

S7 fig. : 25 μg/μl Actinomcyin D (please correct spelling).

Figure S11: Color used in the figure are not very clear. Yellow and gray are clearly explained . Why adding :”other colors indicate genes that did not validate for either rDNA parameter.” Which color (green, purple). Some, but not all background color are explained within figure. Please correct

Legend of S14 Fig. is entitled :”RPL mutants with rDNA organization phenotypes do not have altered level of nascent RNA”, suggesting that more than one RPL mutant will be shown. Within figure, only rpl3001∆ is quantified. Please correct.

Idem for figure S15, mentioning “RPL mutants”, and showing rpl3602∆ and mrt4∆. Please clarify.

Figure S16A, please correct spelling within figure (signaling pathay…)

Reviewer #4: The authors invested a significant amount of work to strengthen the interpretation of their results. Many issues were clearly addressed, but, unfortunately, an important concern remains the lack of a proof that GapR-GFP binds exclusively to transcriptionally induced overtwisted DNA in the different mutants - or physiological conditions. It is convincing that changes in the morphology of the nucleolus occur in the analyzed mutants and conditions, but this observation is not novel unless it is clearly shown that this is due to alteration of the rDNA topology in dependency of the transcriptional state.

1) It is important to convincingly show that GapR-binding correlates with alterations of rDNA topology and not with other expected or unpredictable consequences of nucleolar perturbances. Impairment of ribosome biogenesis by depletion of distinct r-proteins can result in drastic changes of rRNA processing, perturbation of the elongating Pol I machinery and/or R-loop formation. All these could provide a platform for (unspecific) GapR-binding. Analysis of a GapR-mutant that binds DNA nonspecifically cannot rule out these concerns. The authors must show at least one solid proof-of-principal-experiment that the DNA topology in the vicinity of GapR-GFP binding is indeed altered as a consequence of a transition to a different physiological state or in a mutant genetic background.

2) The authors use A43-GFP as rDNA binding protein to underline the GapR-GFP result. A43 is part of the Pol I transcription machinery which is transiently associated with the ribosomal gene and might dissociate when ribosome biogenesis is impaired. It would be more straightforward to use a protein which tightly (constantly) binds to rDNA in areas not transcribed by Pol I. It is not analyzed in this study if r-protein depletion affects Pol I association with the rDNA or the cellular protein level of Pol I subunits.

3) To correlate GapR-binding effects and nucleolar size with Pol I transcription it is important to exclude RNA processing or degradation effects. Dot blot analysis of RNA isolated from cells treated for 5 min with 4-Thio-Uracil presented in this manuscript detects all kinds of RNAs. Depletion of r-proteins can result in rapid degradation of rRNAs and, therefore, dot blot analysis of 4-Thio-Uracil labeled newly synthesized RNA is probably not the best measure for Pol I transcriptional activity. Accordingly, it is important to visualize newly synthesized mature and premature rRNAs. This allows also to evaluate whether Pol I transcription and/or pre-rRNA processing is affected in the mutants and in

**Have all data underlying the figures and results presented in the manuscript been provided?**

Reviewer #1: Yes

Reviewer #2: Yes

Reviewer #4: Yes

PLOS authors have the option to publish the peer review history of their article (what does this mean?). If published, this will include your full peer review and any attached files.

Reviewer #1: No

Reviewer #2: No

Reviewer #4: No

---

## [Editor Report · Decision Letter 2]

4 Jun 2024

Dear Dr Gerton,

Dear Jen,

I am very pleased to let you know that your manuscript has now been accepted for publication in PLOS Genetics.

Thank you for your patience and resilience on this occasion. Your work is data-rich and I am personally convinced it will be greatly appreciated by our community as a substantial piece in this field.

Best wishes,

Denis

Dear Dr Gerton,

We are pleased to inform you that your manuscript entitled "Regulators of rDNA morphology in fission yeast" has been editorially accepted for publication in PLOS Genetics. Congratulations!

Yours sincerely,

Denis LJ LaFontaine, PhD

Guest Editor

PLOS Genetics

Eva Stukenbrock

Section Editor

PLOS Genetics

Comments from the reviewers (if applicable):

**Data Deposition**

http://datadryad.org/submit?journalID=pgenetics&manu=PGENETICS-D-23-00543R2

**Press Queries**

---

## [Editor Report · Acceptance letter]

27 Jun 2024

PGENETICS-D-23-00543R2 

Regulators of rDNA array morphology in fission yeast 

Dear Dr Gerton, 

We are pleased to inform you that your manuscript entitled "Regulators of rDNA array morphology in fission yeast" has been formally accepted for publication in PLOS Genetics! Your manuscript is now with our production department and you will be notified of the publication date in due course.

With kind regards,

Olena Szabo

PLOS Genetics

On behalf of:
